# Learning to Combine Per-Example Solutions for Neural Program Synthesis

**Disha Shrivastava** *
Mila, Université de Montréal
Google Research

**Hugo Larochelle**
Mila, Université de Montréal
Google Research
CIFAR Fellow

**Daniel Tarlow**
Mila, McGill University
Google Research

## Abstract

The goal of program synthesis from examples is to find a computer program that is consistent with a given set of input-output examples. Most learning-based approaches try to find a program that satisfies all examples at once. Our work, by contrast, considers an approach that breaks the problem into two stages: (a) find programs that satisfy only one example, and (b) leverage these *per-example solutions* to yield a program that satisfies all examples. We introduce the *Cross Aggregator* neural network module based on a multi-head attention mechanism that learns to combine the cues present in these per-example solutions to synthesize a global solution. Evaluation across programs of different lengths and under two different experimental settings reveal that when given the same time budget, our technique significantly improves the success rate over PCCoder [30] and other ablation baselines. The code, data and trained models for our work can be found at: https://github.com/shrivastavadisha/N-PEPS.

## 1 Introduction

Program synthesis from examples tackles the problem of coming up with a computer program that satisfies a given set of Input-Output (IO) examples. Since the space of possible programs is large, an exhaustive search can be extremely time-consuming. Therefore, development of systems for program synthesis that can come up with a solution (program satisfying the given IO examples) within a *limited time*, such that it is practical for real-world applications, is a challenging task.

Neural-guided program synthesis systems [4, 30] try to expedite the search by using a neural network conditioned on the IO examples as a learned heuristic for the search procedure. In these systems, a neural network outputs probabilities over programs or properties of programs (e.g. functions). These probabilities are then utilized to guide a search like depth-first or beam search. These systems try to find a program that satisfies all IO examples *simultaneously*, which under most of the settings can be hard. What if instead, we try to find this program in parts? To understand this motivation, imagine a process wherein a programmer is asked to write a program that satisfies a set of unit test cases. They may begin by figuring out a program that satisfies a subset of unit test cases first, and later modifying the program to incorporate other corner cases. Shi et al. [26] uses this intuition to iteratively refine a program by mining fragments of Java code from partial solutions, based on a set of rules and predefined heuristics. Gupta et al. [12] also uses the same intuition, but in a different application for program repair.

In this work, we consider breaking the complex problem of finding a program that satisfies all $N$ given IO examples (called the *global solution*) into $N$ smaller, easy to solve sub-problems, where each sub-problem involves finding a program satisfying only one IO example (called *per-example*

---

*Correspondence to: <dishu.905@gmail.com>

35th Conference on Neural Information Processing Systems (NeurIPS 2021).

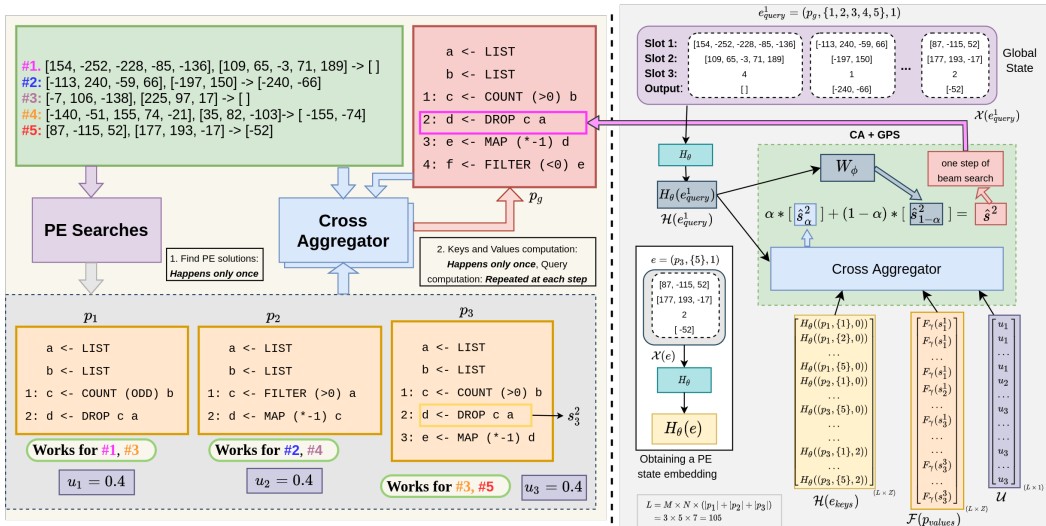

Figure 1: **Idea of N-PEPS:** *(Left)* Illustrating the two stages of N-PEPS with an example; *(Right)* Synthesizing line 2 of $p_g$ using contributions from CA and GPS, with details of how query, keys, values and relation scores are formed. White box shows an example of obtaining a PE state embedding.

*solution*). The cues present in these per-example (PE) solutions are then combined to provide useful signals that can help guide the search for the global solution effectively. As a motivating example, consider the left part of Figure 1, where five IO examples are given as a specification (green box) and we need to find a global solution $p_g$ (red box) that satisfies these five examples. The first stage of our approach consists of performing per-example searches to find a program $p_i$ conditioned on the $i$-th IO example. In our example, we start from IO example #1 and find program $p_1$. In addition, we also check if $p_1$ satisfies any other examples (#3 in figure). Iterating through the examples in this way results in a set of programs $(p_1, p_2, p_3)$ that, taken together, in the ideal scenario, would satisfy all five IO examples. Looking closely at the discovered PE solutions, we see that they contain fragments of the global solution. This brings us to the second stage of our approach that addresses the challenge of how best to aggregate these PE solutions to produce a global solution. Towards that goal, we propose a neural network based architecture, which we refer to as *Cross Aggregator* (CA). It is designed to learn to combine the cues present in these PE solutions, in a way that helps guide the search for $p_g$. We model this aggregation using a multi-head cross-attention mechanism, which leverages the state of step-wise execution of the PE solutions and the synthesized global solution so far (see Section 3.2 for details). Our key contributions can be listed as follows:

- We consider breaking the standard program synthesis pipeline into two stages: (a) discovering PE solutions, and (b) aggregating the PE solutions such that it leads to a global solution. We refer to our approach that uses neural networks at both these stages as *Neural Per-Example Program Synthesis* (N-PEPS).
- We propose a neural network based multi-head attention architecture called Cross Aggregator (CA) that makes use of step-wise execution information to *learn* to combine the PE cues such that it helps guide the search for the global solution.
- We demonstrate via experiments with programs of different lengths and under two different evaluation settings that when given the same time budget, our formulation shows significant improvements in success rate when compared to PCCoder [30] (one of the leading techniques for neural-guided program synthesis) and other ablation baselines.

## 2 Background

Suppose we are given a set $X = \{(x_i, y_i)\}_{i=1}^N = \{r_i\}_{i=1}^N$ of $N$ IO examples and our task is to come up with a program $p_g$ that satisfies these examples. The $i$-th IO example $r_i$ consists of a pair of input $x_i$ and output $y_i$. The program consists of $T$ lines (excluding lines with input variable declarations), i.e. $p_g = [p_g^t]_{t=1}^T$. To be practically meaningful, we impose the constraint that $p_g$ has to be found

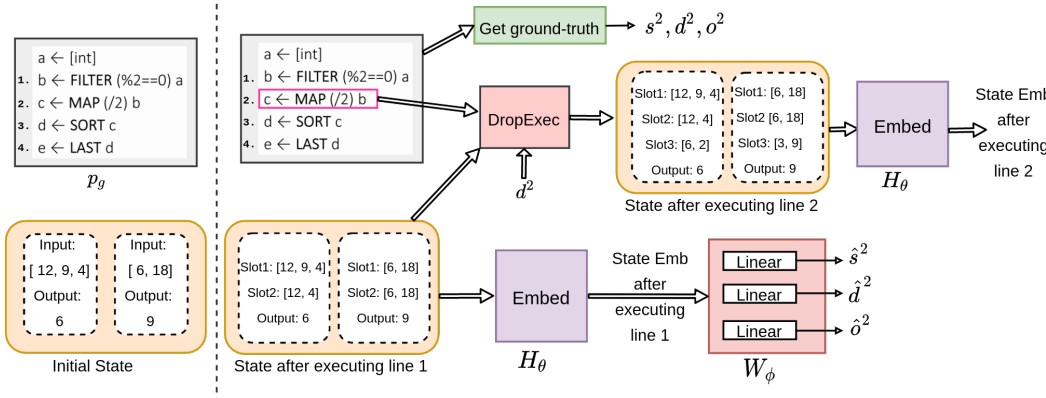

Figure 2: *(Left):* Sample program along with two IO examples that forms the program state at $t = 0$; *(Right):* Block Diagram explaining the training of PCCoder at line 2 of the program.

within a given time budget, specified by a *timeout* value. The syntax and semantics of $p_g$ are governed by a domain-specific language (DSL). We use the DSL provided by Balog et al. [4], which contains first-order functions (e.g. `SORT, REVERSE`) and higher-order functions (e.g. `MAP, FILTER`) that can take *lambda* functions (e.g. (*4), (<0)) as input. The inputs and outputs can be either an integer or a list of integers (see Appendix F of Balog et al. [4] for more details about the DSL). The Predict and Collect Coder (PCCoder) [30] provides state-of-art results for this DSL and is illustrative of methods that directly solve for all available IO examples at once. We refer to these methods as Global Program Search (GPS). We will be building on PCCoder to propose our per-example approach.

## 2.1 PCCoder

PCCoder synthesizes programs one line at a time, through a model based on the notion of a *program state*. The program state is a two-dimensional memory of size $N \times (\nu + 1)$ obtained during the execution of $t$ lines (steps) of a program on a set of $N$ inputs. This means that for each IO example $r_i$, there are up to $\nu$ slots for storing the input and intermediate program variables, with an additional slot for storing the output (see Appendix A.2 for more details). Note that the initial state at $t = 0$ consists of only the IO examples (see left part of Figure 2).

PCCoder consists of two learnable components (i.e. neural networks), $H_\theta$ and $W_\phi$, with parameters $\theta$ and $\phi$. $H_\theta$ obtains the embedding of the current program state by average-pooling the representation of the $\nu + 1$ slots corresponding to individual examples (white boxes inside the state in Figure 2) into a vector of fixed size in $\mathbb{R}^Z$, where $Z$ denotes the embedding size (see Appendix A.2 for details of how these representations of slots are obtained). $W_\phi$ maps this *state embedding* to predictions of three quantities of interest for the next line in the program: (a) the next operator $\hat{o}^t$ (or function e.g. `MAP`); (b) the next statement $\hat{s}^t$ (operator along with its arguments e.g. `MAP(/2) b`); and (c) next drop vector $\hat{d}^t$ which represents positions of variables that can be dropped from the state. The dropping is desirable as it creates slots for storing new variables, which in turn allows for synthesizing longer programs. There is a module called $DropExec$ which executes a given line of the program against an example $r_i$ and stores the resulting variable $c_i$ in the next available slot in the state. If all $\nu$ slots in the state are filled, a variable is dropped from one of the slots using the drop vector and $c_i$ is stored there. The updated state can then be used for predicting the next line (see right part of Figure 2). Next, we provide details of how training and inference is done in PCCoder.

**Training:** For training $H_\theta$ and $W_\phi$, several instances of a specification $X$ and the ground-truth program $p_g$ are provided. Given an instance and line $t$ of the program, training operates by obtaining the ground-truth values of statements ($s^t$), operator ($o^t$) and drop vector ($d^t$). The statement and operator values are represented as one-hot vectors of size equal to the number of statements ($n_s$) and number of operators ($n_o$), respectively in the DSL. The drop vector is a multi-hot vector of size $\nu$ with ones at positions corresponding to the variables in the program that can be dropped, i.e. variables that don't appear in subsequent lines in the program. The step-wise loss $\mathcal{L}$ is the sum of cross-entropy losses between the actual and predicted statement and operator, and the binary cross entropy loss between each position in the actual and predicted drop vector. The task of predicting the

operator is an auxiliary task, i.e. it is used only during training and not at inference time, and is found to improve the training performance. During training, to obtain the updated state, the $DropExec$ module chooses the drop-index to be a random entry from those positions in the drop vector $d^t$ that are ones. The right part of Figure 2 illustrates the process of training at step 2.

**Inference:** Inference is done using complete anytime beam search (CAB) [29] where the time for search is upper bounded by the timeout value. The CAB algorithm operates by performing different beam searches repeatedly in an outer loop. The pruning conditions of the beam search (i.e., beam size, expansion size) are weakened with each iteration of the outer loop, until a solution is found. The inner loop consists of different steps of a single beam search. At each step, the beam consists of the most promising program prefixes, with each prefix represented as a tuple of the current program state, synthesized program until now and the product of the probabilities of the statements in the synthesized program. To synthesize the next line of the program, prefixes are expanded by executing the statements in decreasing order of statement probabilities and taking the argmax of the drop vector probabilities. The statement and drop vector probabilities are obtained using the trained neural networks $H_\theta$ and $W_\phi$. The search terminates if we find a candidate program prefix that satisfies all $N$ IO examples. The corresponding program is the synthesized global solution $p_g$. Note that the search may fail and not discover a global solution within the specified timeout. Appendix A gives details of training and inference procedures, and modules of PCCoder.

# 3   Neural Per-Example Program Synthesis (N-PEPS)

As stated in Section 1, in this work, we decide to break the complex problem of finding a global solution $p_g$ that satisfies all $N$ IO examples, into $N$ smaller sub-problems. Each sub-problem aims to find a program $p_i$ that will satisfy only the IO example $r_i$. The cues present in these PE solutions are then aggregated to help guide the search for $p_g$. We constrain the process of breaking and combining to fit within the specified timeout value. The distribution of total timeout between these stages is treated as a hyperparameter. In this section, we discuss our process of finding PE solutions and follow it with a description of our neural network module that learns to combine the PE solutions.

## 3.1   Per Example Program Synthesis

We refer to the general framework of finding PE solutions first and later aggregating the PE cues to find a global solution, as *Per-Example Program Synthesis* (PEPS). We call the module that finds PE solutions as the *PE Searches* module. To train the PE Searches module, we use the PCCoder model as it is, except that it is trained to take a single IO example as input as opposed to all the examples in $X$. We will call this trained model as the *PE model*. We allocate a fixed value of *PEPS timeout*, which is the maximum time given to find each PE solution. The sum of PEPS timeouts across all PE solutions should be less than the total timeout, so that there is some time left for the CA module to aggregate the PE cues (i.e., $N\times$ PEPS Timeout < Total Timeout). We start from the first example, and using the PE model, try to find a solution that satisfies it. Once found, we also check if this solution satisfies other examples in $X$. We record the fraction of IO examples satisfied by $p_i$, and call it the *PE solution score $u_i$*. If $p_i$ satisfies all examples in $X$ (i.e. $u_i = 1.0$), we stop and return $p_g = p_i$ as the global solution. Otherwise, we proceed to find the next PE solution (based on the order of examples given in $X$). Note that it is possible that for certain examples in $X$, we fail to find a PE solution within the PEPS timeout. Once we have our list of $M$ PE solutions ($0 \le M \le N$), which ideally satisfies all $N$ examples but may not necessarily, we proceed to aggregating them. Note that when comparison with baselines is not a requirement, we can increase speedup by finding PE solutions in parallel (see Appendix D.1 for more details).

## 3.2   Cross Aggregator

**Notation:** To formulate the program state, we define a basic unit called an *execution tuple* (ET). An ET $e = (p, \mathcal{S}, t)$ is a tuple consisting of a program $p$, a subset $\mathcal{S}$ of example indices in $X$ and a step number $t$. Executing the first $t$ steps (lines) of a program $p$ on every example $r_i$ for $i \in \mathcal{S}$ yields a program state which we note as $\mathcal{X}(e)$. Like PCCoder, we pool the representation of slots of the state corresponding to each example $r_i$ for $i \in \mathcal{S}$ to obtain a state embedding (see Section 2.1), hence making its size independent of the size of $\mathcal{S}$. To represent different combinations of programs executed against different sets of examples at different time steps, we define a list **e** of such execution

tuples, with its size denoted by $L$. $(p_1, \{1\}, 0)$ and $(p_3, \{2\}, 2)$ in the bottom right of Figure 1 are examples of such combinations. We then execute each entry in $\mathbf{e}$ to get a list of states $\mathcal{X}(\mathbf{e})$. This is followed by embedding each entry in states $\mathcal{X}(\mathbf{e})$ using $H_\theta$ to yield a tensor of state embeddings $\mathcal{H}(\mathcal{X}(\mathbf{e})) \in \mathbb{R}^{L \times Z}$ (henceforth referred to as $\mathcal{H}(\mathbf{e})$ for simplicity). The white box towards the bottom of Figure 1 shows an example of obtaining a single entry of a PE state embedding.

**Motivation:** To explain the motivation behind CA, let's look at Figure 1, which illustrates the process of synthesizing line 2 of $p_g$. Intuitively, at this step, we will want our aggregation mechanism to have more contribution from line 2 of $p_1$ and $p_3$ (i.e., `DROP c a`). A simple way of aggregating the PE solutions can be to take the sum or mean of the PE one-hot statement vectors (these form our ablation baselines as detailed in Section 4.2). However, this strategy will fail for scenarios that require taking a non-trivial combination of the PE solution statements or cases where the global solution requires the generation of a new statement that is not found in the PE solutions.

In this work, we propose another way of anticipating what line of $p_g$ comes next, that makes use of the execution information of the programs. The idea is to compare the state embedding obtained before executing line 2 of $p_g$ with the PE state embeddings corresponding to each step of execution of the PE solutions. Then, based on the learned relevance of these state embeddings, their corresponding next PE program statements can form valuable cues for synthesizing the next line. In other words, if a particular PE program state has high relevance with the global program state at a given step, then the following PE program line is likely to be useful in synthesizing the next line of $p_g$. We measure this relevance by employing a cross-attention mechanism, with the query formed by the global program state embedding at step $t$, a key formed by the PE program state embedding at step $t$ and the corresponding value formed by the PE program statement at $t+1$. We take a set of such keys and values to form the key matrix $\mathbf{K}$ and the value matrix $\mathbf{V}$, respectively.

**Model:** For synthesizing line $t+1$ of $p_g$, the query $\mathbf{Q}$ is formed from the global state embedding at step $t$, denoted by $\mathcal{H}(\mathbf{e_{query}^t}) \in \mathbb{R}^{1 \times Z}$, where $\mathbf{e_{query}^t} = [(p_g, \{1, 2, \ldots N\}, t)]$. The keys $\mathbf{K} \in \mathbb{R}^{L \times Z}$ are formed from the state embeddings $\mathcal{H}(\mathbf{e_{keys}})$ of the PE solutions. Let $P$ denote the list of $M$ discovered PE solutions, then the list of execution tuples $\mathbf{e_{keys}} = [(p_m, \{j\}, t)]$, where $p_m \in P, j \in \{1, 2, ..N\}, t \in \{0, 1, ..|p_m| - 1\}$, making $L = M \times N \times \sum_{m=1}^{M} |p_m|$. The corresponding PE solution statements form the values $\mathbf{V} \in \mathbb{R}^{L \times Z}$ (more details on how values are obtained is given later). In addition, we have the relation scores $\mathcal{U} \in \mathbb{R}^{L \times 1}$ obtained by taking the PE solution score $u_m$ corresponding to $p_m$ that is part of each ET in $\mathbf{e_{keys}}$. Note that entries in $\mathcal{U}$ are dependent only on the program part in the ET, and independent of the subset of example indices and the time index.

We add position encodings (depending on the time step value of each ET) to $\mathbf{Q}, \mathbf{K}$ and $\mathbf{V}$. This is followed by multiheaded relative attention between our keys, values and query as described in Equation 3. For each head, we perform a scaled dot-product attention [28](Equation 1) and a form of relative attention[2], i.e. taking a mean of the relation scores and attention scores before normalizing with softmax and multiplying with values (Equation 2).

$$Att(\mathbf{Q}, \mathbf{K}) = \frac{\mathbf{Q}\mathbf{K}^T}{\sqrt{d_k}} \tag{1}$$

$$RelAtt(\mathbf{Q}, \mathbf{K}, \mathbf{V}) = \text{softmax}\left(\frac{\mathcal{U}^T + Att(\mathbf{Q}, \mathbf{K})}{2}\right)\mathbf{V} \tag{2}$$

$$MultiHead(\mathbf{Q}, \mathbf{K}, \mathbf{V}) = \text{concat}(head_i, head_2, \ldots head_\tau)W^O \tag{3}$$
$$\text{where} \quad head_i = RelAtt(\mathbf{Q}W_i^Q, \mathbf{K}W_i^K, \mathbf{V}W_i^V)$$

In the equations above, $d_k$ is the dimension of the key, $W_i^Q, W_i^K, W_i^V$ are the query, key and value projection matrices, $\tau$ is the number of heads and $W^O$ is the linear projection that combines the heads. The output from Equation 3 is fed to a positionwise fully-connected feedforward network. We employ a residual connection [13] followed by layer normalization [3] before and after the feedforward network. The resulting encoding is then linearly projected and softmax is applied to get the prediction of the statement for line $t+1$ of $p_g$. We see that our model resembles one layer of the transformer

---

[2]Note that our formulation of relative attention differs from the formulation used in Shaw et al. [25], Hellendoorn et al. [14], where the relation scores are added either to the query or values.

encoder block [28]. Since the keys and query come from different sources, we refer to our model as a *cross* aggregator. Like standard transformers, we can stack multiple blocks of CA. However, since we are operating on a low timeout (5s), we opted for a simple network consisting of only one layer. Details of model parameters can be found in Appendix D.3.

**Obtaining V:** For a key corresponding to an ET consisting of the PE solution $p_m$ and having step index $t$, the value is associated with the statement vector (one-hot vector of size $= n_s$) for step $t + 1$ of $p_m$. Putting together the statement vectors for all execution tuples that are part of $\mathbf{e_{keys}}$, we get a tensor $\mathbf{p_{values}}$ of size $L \times n_s$. Embedding each entry in this tensor using an embedding layer $F_\gamma$ gives us $\mathbf{V} = \mathcal{F}(\mathbf{p_{values}})$ of size $L \times Z$. This is then fed as input to the model described above. The output from the model is then linearly projected to give the logits for the statement predictions $\in \mathbb{R}^{n_s}$ for step $t + 1$ of the global program $p_g$. In addition to the statement predictions, we can also obtain the operator predictions $\in \mathbb{R}^{n_o}$, starting from the operator vector (one-hot vector of size $= n_o$) and following a process similar to the statements, except that we use a different embedding and final projection layer. The right of Figure 1 shows an example of how a query (top) is combined with keys, values and relation scores (bottom) for our model.

## 3.3  Training

The two main components of N-PEPS, the PE Searches module and the Cross-Aggregator, are trained separately. To create samples for training the PE model, we take one data point ($X = \{r_i\}_{i=1}^N$ and $p_g$) from the GPS approach and create $N$ data points out of it. Since we do not have supervision for the PE solutions, for every example $r_i$ in $X$, we use $p_g$ as a proxy for ground-truth PE solution. We believe that using $p_g$ as proxy supervision even though not being entirely correct, forces the PE search component to avoid overfitting to a single example and hence is more likely to produce PE solutions that generalize to examples outside the ones given as specification (see Appendix D.2 for more details).

For training the CA module, we generate data points that we call *aggregator instances*. Each aggregator instance consists of $X$, a list $Y$ of tuples of PE solutions $p_i$ and corresponding PE solution scores $u_i$, and global program $p_g$. The $p_i$'s and $u_i$'s are generated via CAB from a trained PE model (more details on how they are generated in Appendix C.2). Given $X$ and $Y$ as input, the objective is to learn the parameters of the CA module such that the output is the line-wise statement and operator predictions corresponding to $p_g$. The net loss at step $t$ is the sum of two terms: (a) a cross entropy loss between the predicted statement $\hat{s}^t$ (obtained from CA) and the actual statement vector $s^t$ (obtained from $p_g^t$); (b) a cross entropy loss between the predicted operator $\hat{o}^t$ and the actual operator vector $o^t$. Like PCCoder, the operator loss is used as an auxiliary loss to improve training. Note that for each aggregator instance, since we have $X$ and $Y$ to begin with, we need to compute the keys and values only once. However, the computation of query has to be done at each step of the global program execution. While training, since $p_g$ is known, we can use teacher forcing and increase efficiency by batching, where an element in the batch corresponds of one step of execution of $p_g$.

## 3.4  Inference

The process of inference in PEPS is the same as in PCCoder (see Section 2.1), except that in addition to the contribution from GPS, we add another term that accounts for the contribution from CA. The contribution from GPS is obtained by using a *GPS model* that is trained as in standard PCCoder. The net value of the predicted statement at step $t$ is then obtained by taking a weighted contribution from the statement predictions from the trained GPS model $\hat{s}_{1-\alpha}^t$ and the statement prediction from the trained CA module $\hat{s}_\alpha^t$. For predicting the drop vector $\hat{d}^t$, we take contributions only from GPS. When $\alpha = 0$, our approach becomes equivalent to GPS.

$$\hat{s}^t = \alpha * \hat{s}_\alpha^t + (1 - \alpha) * \hat{s}_{1-\alpha}^t \tag{4}$$
$$\hat{d}^t = \hat{d}_{1-\alpha}^t$$

We perform CAB until we find a global solution or we exceed the specified timeout. The right part of Figure 1 illustrates an example of the steps involved in synthesizing step 2 of $p_g$.

# 4 Experiments and Results

Following prior work [4, 30][3], we generate programs for training and testing, with each program consisting of five IO example pairs, i.e., $N = 5$. The data generation process ensures that there is no overlap between the training and test programs, with programs being functionally non-equivalent to programs of shorter or equivalent lengths (see Appendix C.1 for more details). In the first set of experiments (henceforth referred to as **E1**), we generated 105036 training programs of length up to 4 (i.e., consisting of lengths 1, 2, 3, 4). For the second set of experiments (henceforth referred to as **E2**), we generated 189328 training programs of length up to 12. 10% of the training data was used for validation. To ensure robustness and reproducibility of results, for each method, we carry out experiments over 30 different test splits, where each split contains 500 programs of a specific length. For E1, we generate test programs of length 4, and for E2 we generate programs of lengths 5, 8, 10, 12 and 14. We learn separate GPS models and PE models for E1 and E2. All GPS results were obtained using the original PCCoder implementation[3]. A notable difference in our experiments from PCCoder [30] is that we consider a short timeout of 5s (in both E1 and E2, *for all methods*), instead of 5000 and 10000s. This choice is representative of the timeout required for satisfactory user experience in program synthesis systems used in real-world interactive use-cases (such as FlashFill feature in Microsoft Excel [11]). Given a particular timeout value, we record the *Success Ratio*, which is the fraction of test samples that succeeded in finding a global solution.

## 4.1 Initial Experiment: Analysis of PE Solutions

The promise of PEPS is rooted in the assumption that it is much easier to find PE solutions than finding a global solution. In order to test this hypothesis and get an idea of the types of cues discovered by PEPS, we performed a set of analysis experiments using data from E1. Using the trained PE model to find PE solutions, we consider two variants. The first variant called $\mathbf{tot(k)}$ is similar to the strategy of finding PE solutions that we use in PEPS (Section 3.1), where we search for PE solutions sequentially (in the order of examples in $X$) until the discovered PE solutions taken together satisfy $k$ examples in $X$ (where $k \leq 5$). This helps us understand how much the coverage ($= k$) from a list of PE solutions can be. In the second variant called $\mathbf{ind(k)}$, we record success by searching for PE solutions sequentially until we find an individual PE solution that satisfies $k$ out of $N$ examples in $X$. Here, the success ratio helps us assess how good a single PE solution is. In other words, can we rely solely on individual solutions or do we need to aggregate them? For the initial experiment, since no aggregation is done, we divide the timeout of 5s evenly amongst PE searches, i.e., each PE search gets $\frac{1}{5} \times 5 = 1s$ as the timeout value. For GPS, we use the trained GPS model with a timeout of 5s.

Table 1: Success ratio of GPS, $\mathbf{ind}$ and $\mathbf{tot}$ for different values of $k$ for test programs of length 4.

| GPS | ind(1) | ind(2) | ind(3) | ind(4) | ind(5) | tot(1) | tot(2) | tot(3) | tot(4) | tot(5) |
|---|---|---|---|---|---|---|---|---|---|---|
| 77.0 | 99.2 | 95.4 | 85.4 | 70.4 | 43.2 | 99.2 | 97.6 | 97.0 | 94.8 | 82.4 |

Table 1 gives the results of these analysis experiments on one of test splits for programs of length 4. Note that in $\mathbf{tot}$, we are not aggregating the cues to find a global program. Hence, the value given under $\mathbf{tot(5)}$ is not directly comparable to GPS. We make a few observations. First, the success ratio increases with decreasing value of $k$. Therefore, as speculated, it is easier to find solutions that satisfy examples partially. Second, we see that even though the numbers for $\mathbf{ind}$ are encouraging, they are less than the corresponding values (for same $k$) for $\mathbf{tot}$. This suggests that aggregating PE solutions is better than dealing with them individually. Third, the success ratio of $\mathbf{tot(5)}$ is better than GPS. This suggests there is potential in thinking of an architecture that can learn to combine these solutions. Even for cases where we couldn't find PE solutions that satisfy all 5 examples, we can hope to make use of the rich partial cues (indicated by high success ratios) coming from $\mathbf{tot(k < 5)}$.

## 4.2 Methods

In addition to the standard GPS baseline (PCCoder [30]), we experimented with three ablation baselines that represent simple ways of aggregating the PE solutions without making use of the program state. Hence, they help us understand the role of PE cues alone. These baselines are: (i)

---

[3]We used the implementation from PCCoder [30], at https://github.com/amitz25/PCCoder (MIT License) for data generation and obtaining results for PCCoder.

Figure 3: **Results for E1**: *(Left)* Success Ratio with standard error for all models (top row = GPS); *(Center)* Success Ratio vs. time taken; *(Right)* Visualization of attention scores for N-PEPS+$\mathcal{U}$

**Sum-PEPS:** Replacing the contribution from CA module in Equation 4 by a module that combines the PE solutions by taking the sum of all PE one-hot statement vectors; (ii) **Mean-PEPS:** Same as (i) except that sum is replaced by mean; (iii) **Mean-PEPS+$\mathcal{U}$:** Same as (ii) except that the one-hot PE statement vectors are multiplied by their corresponding solution scores before taking the mean. To understand the benefit of aggregating with our proposed CA architecture on top of the value brought by the PE cues, we experimented with the following variations: (i) **N-PEPS:** Our neural model of PEPS described in Section 3.2 with $\mathcal{U}$ being a zero tensor; (ii) **N-PEPS+$\mathcal{U}$:** Same as (i) but with $\mathcal{U}$ included. Complete details of hyperparameters for all methods can be found in Appendix D.

## 4.3 Results

For each test data point, we record either a success or failure (based on whether within 5s, we find a global solution or not) and the actual time taken to find a global solution. As described in Section 3.1, for all PEPS methods, we start by allocating a PEPS timeout value that is less than 1s ($= \frac{1}{N} \times$ total timeout). We sum the actual time ($\leq$ PEPS timeout) taken for finding individual PE solutions. The residual times (if any) left out is added and used for aggregation and global inference. Note that PEPS timeout and $\alpha$ are treated as hyperparameters that are chosen using the validation set.

To provide fair comparison across all methods, each test split is run using a single core and single CPU thread with a timeout of 5s. To account for variability across machines, we chose to run a test split on a machine chosen randomly from a collection of 7 machines of similar configuration (Google Cloud instances with 120GB RAM each)[4]. We report standard error across the 30 test runs.

**Main result:** The left and center parts of Figure 3 show the success ratio and success ratio vs. time taken (average of the actual time taken) plots, respectively, for test programs of length 4 when trained on programs up to length 4 (E1). The left part of Figure 4 shows the success ratio for test programs of lengths 5, 8, 10, 12 and 14, when trained on programs up to length 12 (E2). In both these settings, we observe that performance of the ablation baselines is better than GPS, illustrating the promise in the quality of PE solutions. When we use our CA module to aggregate these cues instead, we see that the performance improves even further. We used the default value of $\nu = 11$ used in PCCoder, which means that for programs of length > 8, certain variables will be dropped from the program state. Also, note that the results for test length 14 represent a case of *length generalization*. We show that in both these scenarios, our proposed method is quite advantageous[5]. In addition, we compare the performance of N-PEPS with GPS for the cases of *intent generalization*, i.e., generalization of the synthesized program to examples other than those given as part of $X$ (Appendix F.2) and when given a longer timeout of 1000s (Appendix F.1). In both these settings, N-PEPS shows superior performance, highlighting its generality.

**Attention visualization:** The right part of Figure 3 shows a visualization of attention scores at $t = 2$ and $t = 3$ obtained from our best model under E1 for the example shown in Figure 1. This example represents a case from the test set where N-PEPS succeeds in finding a global solution, whereas other methods fail. As can be seen, the actual statement of $p_g^2$ is `DROP c a` and our model indeed

---

[4]We additionally verify that different runs on the same machine produce similar results (Appendix D.6)

[5]See Appendix B for success cases of N-PEPS & Appendix F.3 and Appendix F.4 for empirical analysis of synthesized programs.

| Model | Length = 5 | Length = 8 | Length = 10 | Length = 12 | Length=14 |
|---|---|---|---|---|---|
| PCCoder [30] | $70.91 \pm 0.35$ | $44.17 \pm 0.45$ | $28.18 \pm 0.33$ | $19.69 \pm 0.34$ | $14.71 \pm 0.23$ |
| Sum-PEPS | $76.45 \pm 0.33$ | $43.4 \pm 0.56$ | $28.96 \pm 0.27$ | $20.94 \pm 0.32$ | $15.67 \pm 0.32$ |
| Mean-PEPS | $75.79 \pm 0.31$ | $44.42 \pm 0.51$ | $29.55 \pm 0.29$ | $21.45 \pm 0.27$ | $16.35 \pm 0.27$ |
| Mean-PEPS+$\mathcal{U}$ | $75.99 \pm 0.32$ | $44.49 \pm 0.52$ | $29.75 \pm 0.25$ | $21.74 \pm 0.30$ | $16.45 \pm 0.33$ |
| N-PEPS | $79.18 \pm 0.31$ | $\mathbf{47.23 \pm 0.49}$ | $\mathbf{32.3 \pm 0.34}$ | $\mathbf{23.34 \pm 0.28}$ | $\mathbf{17.35 \pm 0.31}$ |
| N-PEPS+$\mathcal{U}$ | $\mathbf{79.19 \pm 0.30}$ | $46.31 \pm 0.61$ | $31.84 \pm 0.36$ | $22.71 \pm 0.28$ | $16.68 \pm 0.21$ |

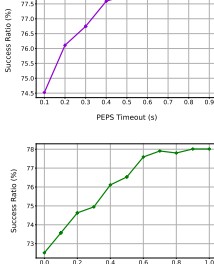

Figure 4: **Results for E2**: *(Left)* Success Ratio with standard error for all models (top row = GPS); *(Right)* Variation of success ratio with PEPS Timeout (top) and $\alpha$ (below) for N-PEPS

gives relatively higher attention scores to $p_1^2$ and $p_3^2$, both of which correspond to the same statement. Similarly at $t = 3$, our model gives more attention to $p_3^3 = $ `MAP (*-1) d` $= p_g^3$.

**Variation with PEPS timeout and $\alpha$**: There is a non-trivial tradeoff in the division of the total timeout into the time given to find PE solutions and the time given to the CA module. A higher PEPS timeout results in better chances of discovering rich PE cues. This means that there may be less or almost no time needed for aggregation. On the other hand, if we start with a low PEPS timeout, the cues from PE solutions may not be as good, but we have more time to perform aggregation. Also, there is a question of how much contribution should be taken from CA and how much from GPS, which is determined by the value of $\alpha$. The right part of Figure 4 analyzes this tradeoff. In the top, we show the variation of success ratio with PEPS timeout and in the bottom, we have the variation with $\alpha$, for the validation set under E2. We see that the performance improves with increase in PEPS timeout with slight decrease towards the end. Also, we see that generally higher values of $\alpha$ are better indicating that the contribution from CA is more important than the contribution from GPS.

**Variants of K:** In addition to obtaining $\mathcal{H}(\mathbf{e_{keys}})$ in the way described in Section 3.2, we tried two other ways of composing the keys by varying the set $\mathcal{S}$ in the execution tuple against which the PE solutions are executed. In the first variant, PE solution $p_m$ from the list $P$ of $M$ discovered PE solutions is executed against the global set $X$, i.e., $\mathbf{e_{keys}} = [(p_m, \{1, 2, ..N\}, t)]$ where $p_m \in P$ and $t \in \{0, 1, ..|p_m| - 1\}$. We denote this variant as **N-PEPS-PG** (PG = PE-global ET). In the second variant, $p_m$ is executed against the set $S_m$ consisting only of examples indices that $p_m$ satisfies, i.e., $\mathbf{e_{keys}} = [(p_m, \{j\}, t)]$ where $j \in S_m$. We call this variant as **N-PEPS-PP** (PP = PE-PE ET). Table 2 shows the test results of these variants for E1 with and without $\mathcal{U}$. We see

Table 2: Performance of variants of **K** for E1

| Variant | Success Ratio |
|---|---|
| N-PEPS-PG | $85.19 \pm 0.26$ |
| N-PEPS-PG+$\mathcal{U}$ | $85.94 \pm 0.26$ |
| N-PEPS-PP | $85.97 \pm 0.26$ |
| N-PEPS-PP+$\mathcal{U}$ | $86.21 \pm 0.27$ |
| N-PEPS | $86.22 \pm 0.25$ |
| N-PEPS+$\mathcal{U}$ | $\mathbf{87.07 \pm 0.28}$ |

that all the variants perform better than GPS and the three ablation baselines (N-PEPS variant used in Section 3.2 was chosen using the validation set). We see similar trend for E2 (see Appendix E).

**New operator discovery by N-PEPS:** We were interested in determining that while synthesizing the global solution how often does N-PEPS rely on copying statements from the PE solutions and how often does it generate new operators. We studied this trend for the cases when $\alpha < 1.0$, i.e., contributions are taken from both CA and GPS as well as when $\alpha = 1.0$, i.e., contributions are taken only from CA (Appendix G.2). This question is important as it helps us understand the generalization capabilities of CA outside the statements in the PE solutions. We found that CA alone (with $\alpha = 1.0$) is capable of generating new operators. In addition, we found that the new operators are present as part of the nearest neighbours of the PE statements, thereby pointing to an increased likelihood of these being ranked higher in the beam search and hence being present in the global solution (see Appendix G.3 and G.4 for details).

## 5 Related Work

There have been numerous efforts on using deep learning for program synthesis [4, 6, 9, 15, 8, 17–19, 21]. However, there is less work that uses the execution of partial programs to assist in synthesis.

PCCoder [30] is one such work, which we describe in Section 2.1. BUSTLE [20] reweighs the sub-expressions in bottom-up program synthesis using the intermediate values obtained by execution of sub-expressions along with property signatures. REPL [10] executes partial programs using learned policy and value networks. Chen et al. [7] uses a neural encoder-decoder architecture to generate program tokens conditioned on intermediate states obtained from execution of partial programs. They work with the Karel DSL [22, 6] that contains loops and conditionals, an attribute missing from the DSL which we work with. Therefore, extending N-PEPS for Karel is an interesting future work. Note that all the approaches mentioned above are examples of purely GPS approaches.

Few works use solutions that satisfy examples partially, to aid in program synthesis. Initial motivation for our work comes from FrAngel [26], which is a component-wise synthesis system that relies on mining fragments of Java code that satisfy examples partially, given target program function signatures and a list of Java libraries. The mining of fragments as well as combination is done using a set of heuristics and predefined rules with no deep learning involved. Assuming that the user provides IO examples in order of increasing difficulty, Perelman et al. [24] iteratively refines a program, with the current program satisfying the sequence of IO examples encountered till now. STUN [1] extends the CEGIS [27] approach by providing domain-specific explicit unification operators for combining partial solutions while Alur et al. [2] uses decision trees for the same. Recently, BESTER [23] and later PROBE [5] perform bottom-up enumeration of programs in a loop by enumerating all programs that satisfy IO examples partially. This is followed by heuristics-based selection of promising programs. However, as opposed to N-PEPS that automatically learns to aggregate partial solutions producing the global program in one shot, PROBE relies on using these programs to iteratively update the weights of useful productions in their probabilistic grammar using a fixed update rule. This update can be viewed similar to our ablation baselines that do not use the neural network based learned aggregation. The guided-search component of PROBE provides an interesting alternative to finding PE solutions. One way of incorporating this in our top-down setting might be to start with the CAB search as in GPS and then select promising solutions based on evaluating examples on prefixes of programs obtained during the beam search. It may be useful to then aggregate the selected solutions using a neural architecture similar to ours.

## 6    Conclusions and Future Directions

In this work, we propose N-PEPS, where the idea is to break the problem of finding a program that solves all examples into two stages: (a) finding programs that solve a single example (PE solutions) (b) aggregating the PE solutions such that it leads to a program that solves all examples. For aggregation, we propose a neural-network based multi-head attention architecture (CA module) that utilizes the state of program execution to learn to combine the PE cues. We note that program synthesis systems in general should be deployed with caution for use in real-world applications. Blind trust on these systems can create chances for potential negative impact, as there might be cases where the generated program contains bugs, especially for unseen examples outside the specification. In the future, we want to work with programs that contain loops and conditionals [22, 7, 6]. Another interesting research direction would be to explore the interaction of N-PEPS with out-of-distribution generalization settings like compositional generalization [16].

## Acknowledgments and Disclosure of Funding

Hugo Larochelle would like to acknowledge the support of Canada Research Chairs and CIFAR for research funding. The authors would like to thank Google Cloud for providing compute resources required for this project. We would like to thank the anonymous reviewers for their valuable comments and thorough engagement during the rebuttal phase that helped us improve our paper. We would also like to extend our vote of thanks to Daniel Johnson, Petros Maniatis, Jacob Austin, Koushik Sen, David Bieber, Sandeep Subramanian, Varsha Embar, Nicolas Gontier and Andreea Deac for feedback and comments on the draft that helped us improve writing. We would also like to acknowledge the quick and helpful response from Amit Zohar and Lior Wolf on queries related to the PCCoder implementation.

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
