# A    Details of PCCoder [5]

We provide our version of the training and inference algorithms and description of modules used in PCCoder [5] next. Note that the terminology used in PCCoder [5] differs from what we have used here.

## A.1    Training and Inference Algorithms of PCCoder

**Algorithm 1** Train (GPS)

**Require:** $p_g = [p_g^t]_{t=1}^T$ = ground-truth program with $T$ lines
**Require:** $\nu$ = max # allowed variables = memory-size
**Require:** $X = \{(x_i, y_i)\}_{i=1}^N = \{r_i\}_{i=1}^N$ = set of $N$ IO examples

1: $\mathcal{X}^0 = [r_i]_{i=1}^n$ ▷ *Initial State*
2: **for** $t$ in $range(T)$) **do**
3:      ▷ *Obtain ground truth*
4:      $s^t, o^t, d^t = R(p_g^t, [p_g^j]_{j \geq t}, \mathcal{H}^{t-1})$
5:      $\mathcal{H}^{t-1} = H_\theta(\mathcal{X}^{t-1})$ ▷ *Obtain current state embedding*
6:      $\hat{s}^t, \hat{o}^t, \hat{d}^t = W_\phi(\mathcal{H}^{t-1})$ ▷ *Obtain predictions*
7:      ▷ *Calculate loss and update parameters*
8:      $\mathcal{L} = \text{CE}(s^t, \hat{s}^t) + \text{CE}(o^t, \hat{o}^t) + \Sigma_{j=1}^\nu \text{BCE}(d^{t^j}, \hat{d}^{t^j})$
9:      $\theta \leftarrow \theta - \alpha * \nabla_\theta \mathcal{L}$
10:      $\phi \leftarrow \phi - \alpha * \nabla_\phi \mathcal{L}$
11:      ▷ *Randomly chose an index to drop*
12:      $d'^t = \text{random\_choice}(d^t)$
13:      ▷ *Execute $p_g^t$ to get updated state*
14:      $\mathcal{X}^t = DropExec(p_g^t, \mathcal{X}^{t-1}, d'^t, \nu)$

15: **procedure** DROPEXEC$(p, x, d', \nu)$
16:      $l = get\_num\_vars(x)$
17:      $N = shape(x)[0]$ ▷ *# IO examples*
18:      **for** $i$ in range($N$) **do**
19:          $x_i = x[i]$
20:          ▷ *Execute $p$ against $x_i$ to obtain result $c_i$*
21:          $c_i = Execute(p, x_i)$
22:          **if** $l > \nu$ **then** ▷ *Need to drop a variable*
23:              $x_i[d'] = c_i$
24:          **else**
25:              $x_i.append(c_i)$
26:      $l = l + 1$
27:      $set\_num\_vars(x, l)$
28:      **return** $x$ ▷ *return the updated state*

**Algorithm 2** Inference (GPS)

1: $\mathcal{X}^0 = [r_i]_{i=1}^n$
2: **while** time < timeout **do** ▷ CAB outer loop
3:      ▷ *Initial Beam: (state, program, prob)*
4:      $B = [(\mathcal{X}^0, [\ \ ], 1.0)]$
5:      ▷ *$p_g$ = global solution*
6:      $p_g = BeamSearch(B)$ ▷ CAB inner loop
7:      **if** $p_g$ == FAILED **then**
8:          beam_size*=2; beam_expansion_size+=10

9:      **procedure** BEAMSEARCH$(B)$
10:          **while** beam search conditions are met **do**
11:              $B' = [\ ]$ ▷ *new beams*
12:              ▷ *For each parent node*
13:              **for** $(b, (\mathcal{X}_b^{t-1}, p_b^{t-1}, s_b^{t-1}))$ in enum($B$) **do**
14:                  **if** is_solution($\mathcal{X}_b^{t-1}$) **then**
15:                      **return** $p_b^{t-1}$
16:                  $\mathcal{H}_b^{t-1} = H_\theta(\mathcal{X}_b^{t-1})$
17:                  $\hat{s}_b^t, \hat{d}_b^t = W_\phi(\mathcal{H}_b^{t-1})$
18:                  ▷ *sort $\hat{s}_b^t$ by decreasing probability*
19:                  $\hat{s}_b^t = sort(\hat{s}_b^t)$
20:                  ▷ *choose argmax of $\hat{d}_b^t$ to drop*
21:                  $d'^t_b = argmax(\hat{d}_b^t)$
22:                  ▷ *Expand the parent node*
23:                  **for** $\tilde{s}_b^t$ in $\hat{s}_b^t[: expansion\_size]$ **do**
24:                      ▷ *get statement id for the prob entry*
25:                      $p_b^t = prob\_to\_stat(\tilde{s}_b^t)$
26:                      ▷ *get updated memory*
27:                      $\mathcal{X}_b^t = DropExec(p_b^t, \mathcal{X}_b^{t-1}, d'^t_b, \nu)$
28:                      $p_b^{t-1}.append(p_b^t)$ ▷ *updated program*
29:                      $s_b^{t-1} = s_b^{t-1} * \tilde{s}_b^t$ ▷ *updated probability*
30:                      $B'.append((\mathcal{X}_b^t, p_b^{t-1}, s_b^{t-1}))$
31:              ▷ *sort beams by decreasing probability*
32:              $B' = sort(B')[:beam\_size]$
33:              $B = B'$
34:          **return** FAILED ▷ *if no solution found during beam search, return Failed solution*

## A.2    Description of Modules in PCCoder

The inputs to a program can either be an integer or an array of integers of maximum length 20. The integers can be in the range [-256, 255]. There can be a maximum of three input arguments to a program. There are 1298 statements and 38 operators in the DSL, i.e., $n_s = 1298$ and $n_o = 38$. Execution of a line in the program returns exactly one variable. Below, we describe the blocks present at different stages of PCCoder:

- **State Representation:** For each of the $N$ IO examples, the corresponding inputs are taken and all entries are made positive by subtracting the minimum integer value under the DSL (i.e. -256) from them. For shorter inputs, NULL values up to length $q = 20$ are padded. Then two bits indicating the type of input (list or int) are appended at the beginning of this representation. Therefore, each variable is now represented as a vector of size $q + 2 = 22$. There can be a maximum of $\nu$ input variables and one output variable (corresponding to the output of the IO example given). If there are less than $\nu$ variables, NULL values are padded to make it uniform. An account of the actual number of variables (i.e. number of filled slots) present in the state is also kept, denoted by $l$. The output of this stage is an array of size $N \times (\nu + 1) \times (q + 2)$. This forms the *state* $\mathcal{X}$.

- **State Embedding ($H_\theta$):** The output obtained in the previous step is then passed through a series of neural network blocks to obtain a *state embedding* $\mathcal{H}$. An embedding layer projects each entry in the state (excluding the type bits) into an vector of size $e = 20$, giving us a tensor of size $N \times (\nu + 1) \times (q * e + 2)$. This is then passed through a linear layer of size 56 and then reshaped to obtain a tensor of size $N \times (\nu + 1) * 56$. It is then passed through a dense block to obtain a tensor of size $N \times Z$ where $Z = 256$. This pre-pooling version is what we refer to as the representation of slots in Section 3.2. An average pooling of these representations across all $N$ examples gives a vector of size $1 \times 256$ that forms the state embedding.
- **Predicting quantities of interest in next line ($W_\phi$):** The state embedding obtained above is projected into three linear heads of size 1298, 38 and $\nu$ followed by softmax, softmax and sigmoid, respectively which gets us the statement, operator and drop probabilities.
- **DropExec:** In the $DropExec$ module, after a statement is executed against the variables present in the slots in the state $\mathcal{X}^0$, we get new values of resulting variables. If the actual number of variables $l$ exceeds $\nu$, one of the existing variables is dropped based on the drop vector. If not, this new variable is simply appended to the existing variables by filling the next slot in the memory. This updated state is then passed through $H_\theta$ to get the updated state embedding $\mathcal{H}^1$. This completes one step of execution of the program.

For the next steps, we repeat the last two steps mentioned above till we reach the end of the program. See Figure 2 for an illustration of the process at $t = 2$.

## B Sample Cases

Below we provide two sample cases where GPS fails and our N-PEPS model (for E2) succeeds in finding a global solution. Foe each sample case, we show the synthesized global solution on the left, the set of IO examples in the center and the discovered PE solutions along with PE solution scores in the right. We also report the actual time taken to find the solutions. Note that for the second case, even though the global ground-truth test program is of length 8, N-PEPS discovers a global solution of shorter length.

| Global Solution: | IO examples: | PE Solutions: |
|---|---|---|
| (Time taken to find=3.21s) | **#1.** *Input*: [4, 5, 6, 2, 6, 2, 1, 6, 1, 4, 2, 5, 6, 3, 2, 2] | $\mathbf{p_1}$ : Time taken to find=0.2s Satisfies #1, #4 ($u_1 = 0.2$) |
| | *Output*: [4, 12, 10, 8] | |
| a ← LIST | **#2.** *Input*: [3, 2, 5, 0, 3, 2, 3, 0, 4, 1, 0, 2, 3, 0, 3, 4] | a ← LIST |
| b ← ZIPWITH (+) a a | | b ← ZIPWITH (+) a a |
| c ← TAIL b | *Output*: [6, 0, 10, 4, 6] | c ← TAIL b |
| d ← TAKE c b | **#3.** *Input*: [1, 1, 4, 0, 0, 0, 0, 5, 0, 5, 3, 5] | d ← TAKE c b |
| e ← COUNT (>0) d | | e ← REVERSE d |
| f ← TAKE e d | *Output*: [2, 2] | |
| g ← COUNT (>0) f | **#4.** *Input*: [4, 4, 1, 4, 4, 1, 4, 2, 2, 1, 3, 4] | $\mathbf{p_2}$ : Time taken to find=0.34s Satisfies #2, #3, #4, #5 ($u_2 = 0.8$) |
| h ← TAKE g f | | |
| i ← TAKE g h | *Output*: [4, 8, 2, 8, 8, 2, 8, 8] | a ← LIST |
| j ← HEAD i | **#5.** *Input*: [4, 1, 1,, 3, 3, 1, 4, 0, 4, 2, 4] | b ← ZIPWITH (+) a a |
| k ← TAKE j i | | c ← HEAD b |
| l ← TAKE j k | *Output*: [8, 2, 6, 6, 2, 2, 8] | d ← TAKE c b |
| m ← TAKE j k | | e ← COUNT (>0) d |
| n ← TAKE j k | | f ← TAKE e d |
| o ← REVERSE n | | g ← REVERSE f |

**Global Solution:**
(Time taken to find=2.98s)

```
a ← LIST
b ← INT
c ← MAXIMUM a
d ← TAKE c a
e ← TAIL c
f ← TAKE b c
g ← ZIPWITH (+) f f
h ← MAP (+1) g
i ← TAKE e h
```

**IO examples:**
**#1.** *Input*:
[1, 0, 3, 3, 3], 35
*Output*:
[3, 1, 7]
**#2.** *Input*:
[6, 3, 3, 1, 2, 2, 0, 3, 8, 7], 50
*Output*:
[13, 7, 7]
**#3.** *Input*:
[1, 5, 6, 10, 5, 11, 7, 0, 7, 11, 10, 9, 4], 78
*Output*:
[3, 11, 13, 21, 11, 23, 15, 1, 15, 23]
**#4.** *Input*:
[12, 4, 11, 11, 4, 7, 12, 11, 11, 10, 5, 8, 9, 8], 166
*Output*:
[25, 9, 23, 23, 9, 15, 25, 23]
**#5.** *Input*:
[4, 0, 5, 5, 1, 1, 1, 1], 126
*Output*:
[9]

**PE Solutions:**
$\mathbf{p_1}$ : Time taken to find=0.17s
Satisfies #1, #4, #5 ($u_1 = 0.6$)

```
a ← LIST
b ← INT
c ← TAIL a
d ← TAKE c a
e ← ZIPWITH (+) d d
f ← MAP (+1) e
```

$\mathbf{p_2}$ : Time taken to find=0.37s
Satisfies #1, #5 ($u_2 = 0.4$)

```
a ← LIST
b ← INT
c ← TAKE b a
d ← TAIL c
e ← ACCESS d c
f ← TAKE e c
g ← ZIPWITH (+) f f
h ← MAP (+1) g
```

$\mathbf{p_3}$ : Time taken to find=0.8s
Satisfies None ($u_3 = 0.0$)

```
FAILED
```

$\mathbf{p_4}$ : Time taken to find=0.17s
Satisfies #1, #4, #5 ($u_4 = 0.6$)

```
a ← LIST
b ← INT
c ← TAIL a
d ← TAKE c a
e ← ZIPWITH (+) d d
f ← MAP (+1) e
```

$\mathbf{p_5}$ : Time taken to find=0.17s
Satisfies #1, #4, #5 ($u_5 = 0.6$)

```
a ← LIST
b ← INT
c ← TAIL a
d ← TAKE c a
e ← ZIPWITH (+) d d
f ← MAP (+1) e
```

# C   Data Generation

## C.1   Generation of Training and Test set

Similar to the data generation process described in Balog et al. [1], Zohar and Wolf [5] and using the implementation from PCCoder [1], we generated programs for training and testing where each program consists of five input-output examples. The process starts by generating training programs iteratively starting from length 1 till the maximum length specified (4 and 12 in our case). For each length, first a program of that length is generated followed by generating corresponding IO examples which correspond to that program. This is followed by checking for functional non-equivalence of that program with all generated programs so far (i.e., programs of length less than or equal to the current length). Functional non-equivalence means that given a set of IO examples, we can't have a program of length x that satisfies the set of examples when we already have a program of length less than or equal to x in our dataset that satisfies the same set of examples. If the program is found functionally equivalent to any other programs, it is discarded, else it is added to the training set.

Once the generation of training set is complete, we proceed to generating the test set. Given a test length, we generate a program of that length followed by generating the corresponding IO example pair. In addition to checking for functional non-equivalence with all programs in the test set so far, we also test for functional non-equivalence with every program in the training set. This makes sure that there there is no overlap between the training and test sets and all the programs are functionally non-equivalent to each other. We have two experimental settings: (a) **E1:** Training set = 105036 programs till length 4 and 30 test sets of 500 programs each of length = 4; (b) **E2:** Training set = 189328 programs of length up to 12 and 30 test sets of 500 programs each of lengths = 5, 8, 10, 12 and 14. In each setting, 10% of the training data was used for validation. Figure 1 shows the distribution of training programs with length in both the settings. There are less programs of longer lengths as there is high probability that they end up being discarded because a functionally equivalent program of shorter length was found.

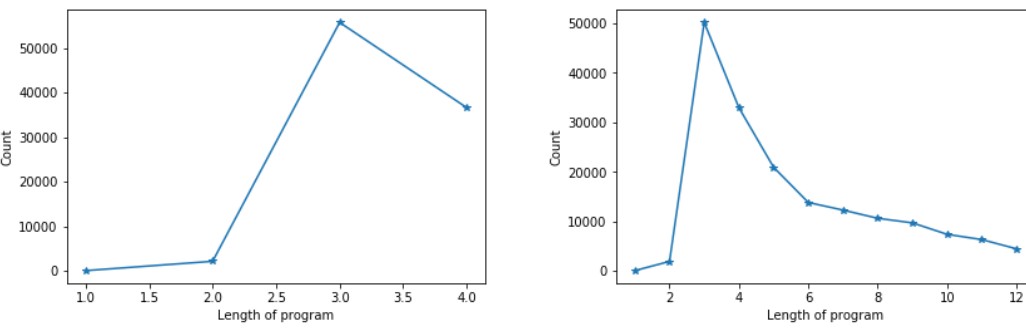

Figure 1: **Distribution of training programs**: *(Left)* For E1; *(Right)* For E2

## C.2   Generation of Aggregator Instances

An aggregator instance consists of the set of IO examples $X$, a list $Y$ of PE solutions $p_i$ along with the corresponding PE solution scores $u_i$, and the corresponding global program $p_g$. To create aggregator instances, for each data point (given $X$ and $p_g$) in the original training dataset (generated as described in C.1), we generate PE solutions and PE solution scores using the PE Searches module. For generating the PE solutions, we need to choose a value of PEPS timeout. We generated aggregator instances with PE solutions obtained using the trained PE model, in three ways: (a) one aggregator instance with a fixed PEPS timeout of 0.5s; (b) two aggregator instances with PEPS timeout randomly chosen from [0.1s, 0.2s, .. 0.9s]; (c) three aggregator instances each with PEPS timeouts of 0.4s, 0.5s and 0.6s, respectively. These options will lead to the same, twice and thrice the number of data points present in the original training set. We chose to omit a sample from being part of training data formed from aggregator instances if either (a) An aggregator instance consists of a PE solution that satisfies all examples (i.e., $u_i = 1.0$) or (b) When we fail to get any PE solution (i.e., all $u_i = 0.0$). We can then generate data which omits both (a) and (b). The datasets formed after removing these

---

[1]https://github.com/amitz25/PCCoder (MIT License)

Table 1: Aggregator data statistics for E1 and E2

| Dataset | # of samples | |
| --- | --- | --- |
| | E1 | E2 |
| $D_{0.5}^{train}$ | 16408 | 82102 |
| $D_{0.5}^{val}$ | 2132 | 9492 |
| $D_{rand}^{train}$ | 41707 | 176235 |
| $D_{rand}^{val}$ | 5365 | 20035 |
| $D_{0.5\pm0.1}^{train}$ | 49116 | 248972 |
| $D_{0.5\pm0.1}^{val}$ | 6396 | 28734 |

aggregator instances will be referred to as $\mathbf{D_{0.5}}$, $\mathbf{D_{rand}}$ and $\mathbf{D_{0.5\pm0.1}}$ for cases (a), (b) and (c), respectively. In addition, within each aggregator instance, we can chose to discover all 5 PE solutions (***all***) or alternatively find a list of $M$ PE solutions where $M \leq 5$ such that taken together these satisfy all examples in $X$ (***tot***). Therefore, in total we have 12 different variations (3 PEPS timeouts $\times$ 2 inclusion conditions $\times$ 2 modes of discovering PE solutions) of training datasets which can be used for training the cross-aggregator. We follow a similar procedure to generate variations of corresponding validation datasets which are used to select the hyperparameters and early-stopping for training with each dataset variation. Table 1 gives the training and validation data statistics (note that the 2 modes for discovering PE solutions will affect the content of a single aggregator instance, but the number of aggregator instances will remain the same in both cases).

## D Experimental details

All our implementations are based on PyTorch [3] version 1.4.0. The training for the GPS and PE models and the CA was done using Tesla V100 (16GB) and Tesla P100 (16GB) GPUs on Google Cloud instances with 120GB RAM.

### D.1 Parallel Execution for PEPS

In the current formulation, we find PE solutions sequentially. However, the running time can be reduced further by finding PE solution in parallel as the process of finding PE solution $i$ is dependent only on the IO example $r_i$. So, instead of finding PE solutions one by one, we can find PE solutions for all examples in parallel and then check whether the PE solution $p_i$ satisfy any other example from $X$ apart from $r_i$. The total time for PEPS can then be thought of max(time taken to find a PE solution that satisfies $r_i$) + time taken to aggregate. However, one could argue that PCCoder can also employ more threads in parallel to speed up their search. Therefore, for a fair comparison with PCCoder, we decided to find PE solutions sequentially where we evaluate both N-PEPS and PCCoder on a single CPU thread (with no parallel computations). However, when being deployed for an application where comparisons with other methods are not required, N-PEPS can significantly boost the speed up by searching for PE solutions in parallel in a way suggested above.

### D.2 Training the GPS and PE models

For each experimental setting, we used the training set (generated as described in C.1) as it is for training the GPS model. For training the PE model, we created five entries out of a single training data point such that a modified entry has a single IO example and the corresponding program is the same across all five entries = program in the data point for GPS. Since we don't have supervision available for PE solutions, we chose $p_g$ to serve as a proxy for ground-truth of these PE solutions. Another way of creating this supervision would have been to perform separate PE searches for each example and recording the discovered PE solution as ground-truth. However, this procedure would have required the selection of a specific PE timeout. We didn't have any good idea of how to select this value as it would have influenced the generated PE solutions, hence the supervision itself. Also,

we didn't know what would have been the best supervision to use for cases where the PE search fails to find a solution. We believe that using $p_g$ as proxy supervision even though not being entirely correct, forces the PE search component to avoid overfitting to a single example and hence is more likely to produce PE solutions that exhibit intent generalization (generalization to examples outside the ones given as specification).

The number of training points for PE model = 5 * number of training data points for GPS. The corresponding validation split was used to select hyperparameters. The selected hyperparameter values were:

- *GPS model:* learning rate = 0.001; batch size = 32 for E1 and 100 for E2.
- *PE model:* learning rate = 0.001; batch size = 100 for E1 and 256 for E2.

For both settings $\nu = 11$. This means that the state has slot for storing 7 intermediate variables, 3 slots for input variables (there can be a maximum of 3 input arguments to a program) and an additional slot for storing the output. This means that for E2, dropping will happen for programs of length greater than 8. We used Adam [2] optimizer with a learning rate scheduler that decayed the learning rate by 0.1 every 4 epochs. We used the validation set for early-stopping. Let's call the learned PE modules as $H_{\theta_{pe}}$ and $W_{\phi_{pe}}$ whereas, the corresponding GPS modules to be $H_{\theta_g}$ and $W_{\phi_g}$.

### D.3 Training the Cross Aggregator

For both E1 and E2, we train our cross aggregator (CA) module using the variants of keys mentioned in Section 3.2 and Section 4.3. For N-PEPS-PG, we use $H_{\theta_g}$ to obtain state embeddings that forms the keys, whereas for N-PEPS and N-PEPS-PP we used $H_{\theta_{pe}}$. For faster convergence, we initialize the statement and operator heads with the corresponding statement and operator linear heads from $W_{\phi_g}$. We tried finetuning the parameters of $H$, but it didn't result in significant difference in training performance. Hence, we decided to leave the parameters of the $H$ module unaltered during training. As mentioned in C.2, we tried both *all* and *tot* ways of discovering PE solutions while training. In equations 1, 2 and 3 in Section 3.2, the projection matrices $W_i^Q \in \mathbb{R}^{d_{model} \times d_q}$, $W_i^K \in \mathbb{R}^{d_{model} \times d_k}$, $W_i^V \in \mathbb{R}^{d_{model} \times d_v}$, $W^O \in \mathbb{R}^{\tau d_v \times d_{model}}$. For the multihead relative attention, we used $d_k = d_q = d_v = 64$, $\tau = 8$ and $d_{model} = 256$. A dropout value of 0.1 was used while training.

### D.4 Details of Training Hyperparameters

We tried different values of learning rates, optimizers, learning rate schedulers, datasets and the PE discovery options. We tried three types of learning rate schedulers[2]: (a) **cosine**: `torch.optim.lr_scheduler.CosineAnnealingLR(optimizer, T_max=10, eta_min=0)` ;
(b) **cosinewarm**: `torch.optim.lr_scheduler.CosineAnnealingWarmRestarts(optimizer, T_0=10)`
; (c) **reduceonplateau**: `torch.optim.lr_scheduler.ReduceLROnPlateau(optimizer, 'min')`
where `optimizer = Adam, SGD`. Below we provide the hyperparameter configuration for the best models chosen using the validation set.

### D.5 Details of Inference Hyperparameters

For inference we use CAB [4] which consists of performing beam search iteratively, with pruning conditions of beam search (i.e., beam size, expansion size, etc.) weakened with each iteration, until a solution is found. Simialr to PCCoder [5], we start with beam size = 100, expansion size = 10 and maximum depth of beam search = number of steps = maximum program length. If the beam search fails, we double the beam size and increase the expansion size by 10, and perform beam search again with the modified parameters. The beam search terminates if we exceed the timeout. If no solution is found at the end of CAB, we mark that solution as FAILED.

We created a smaller validation split called *smallval* which consists of 5% of the samples chosen randomly from the larger validation data. The size of smallval is 525 samples and 946 samples for E1 and E2, respectively. We used this set to find optimal values of PEPS timeout and $\alpha$ for each model.

---

[2]see https://pytorch.org/docs/stable/optim.html for more details

Table 2: Hyperparameter values for training the CA. lr= learning rate, lrs = learning rate scheduler, o=optimizer

| Model | Hyperparameters | |
|---|---|---|
| | E1 | E2 |
| N-PEPS-PP | $D_{0.5}$, *all*, lr=1e-4, o =Adam, lrs=cosine | $D_{0.5\pm0.1}$, *tot*, lr=1e-4, o=Adam, lrs=reduceonplateau |
| N-PEPS-PP+$\mathcal{U}$ | $D_{rand}$, *all*, lr=1e-4, o=SGD, lrs=cosinewarm | $D_{rand}$, *all*, lr=1e-4, o=SGD, lrs=cosinewarm |
| N-PEPS-PG | $D_{rand}$, *tot*, lr=1e-4, o=SGD, lrs=cosine | $D_{0.5}$, *all*, lr=1e-4, o=SGD, lrs=cosine |
| N-PEPS-PG+$\mathcal{U}$ | $D_{rand}$, *all*, lr=1e-4, o=SGD, lrs=cosinewarm | $D_{0.5}$, *all*, lr=1e-4, o=Adam, lrs=reduceonplateau |
| N-PEPS | $D_{rand}$, *all*, lr=1e-4, o=SGD, lrs=cosine | $D_{rand}$, *all*, lr=3e-4, o=Adam, lrs=cosine |
| N-PEPS+$\mathcal{U}$ | $D_{0.5}$, *tot*, lr=3e-4, o=Adam, lrs=cosinewarm | $D_{rand}$, *all*, lr=1e-4, o=Adam, lrs=reduceonplateau |

Table 3 provides the selected hyperparameter values for all the models in both the settings. Timeout of 5s is divided between PE Searches module and the aggregation + GPS module. The time allocated to latter is denoted by GT in the table. For GPS, since no PE solutions are discovered, the whole timeout is allocated to the GPS inference block and no aggregation happens, i.e., $\alpha = 0.0$.

Table 3: Hyperparameter values for Inference. PT = PEPS timeout, GT = 5 - ( 5 * PT ).

| Model | Hyperparameters | |
|---|---|---|
| | E1 | E2 |
| GPS | GT=5.0s, PT=0.0s, $\alpha$=0.0 | GT=5.0s, PT=0s, $\alpha$=0.0 |
| Sum | GT=2.5s, PT=0.5s, $\alpha$=0.8 | GT=0.5s, PT=0.9s, $\alpha$=0.2 |
| Mean | GT=2.5s, PT=0.5s, $\alpha$=0.8 | GT=0.5s, PT=0.9s, $\alpha$=0.2 |
| Mean+$\mathcal{U}$ | GT=2.5s, PT=0.5s, $\alpha$=0.9 | GT=0.5s, PT=0.9s, $\alpha$=0.4 |
| N-PEPS-PP | GT=1.0s, PT=0.8s, $\alpha$=0.8 | GT=1.5s, PT=0.7s, $\alpha$=0.8 |
| N-PEPS-PP+$\mathcal{U}$ | GT=1.0s, PT=0.8s, $\alpha$=0.7 | GT=2.5s, PT=0.5s, $\alpha$=1.0 |
| N-PEPS-PG | GT=1.0s, PT=0.8s, $\alpha$=0.8 | GT=2.0s, PT=0.6s, $\alpha$=0.9 |
| N-PEPS-PG+$\mathcal{U}$ | GT=1.0s, PT=0.8s, $\alpha$=0.8 | GT=1.0s, PT=0.8s, $\alpha$=1.0 |
| N-PEPS | GT=0.5s, PT=0.9s, $\alpha$=0.8 | GT=1.0s, PT=0.8s, $\alpha$=0.8 |
| N-PEPS+$\mathcal{U}$ | GT=1.0s, PT=0.8s, $\alpha$=0.8 | GT=2.0s, PT=0.6s, $\alpha$=0.9 |

### D.6 Variation across different runs and machines

To ensure robustness and reproducibility of our results, we performed experiments with variations along three dimensions: different runs on the same machine, different machines and different test splits. Table 4 presents the results of variation in success ratio for E1 when run across different test splits, machines and runs across a single machine. Each run consists of a single CPU thread and single core setting on a machine (Google Cloud instance with 120GB RAM). We can see that there is very little variation for runs across the same machine. Hence, for our main experiments we chose to report standard error across different test splits with single runs on machines that are chosen randomly from a pool of 7 Google Cloud instances with same configuration.

## E  Results for Variants of Key for E2

Table 5 presents the test results of ablation studies with different variants of keys for E2. Similar to E1, we see that all the variants perform better than the corresponding values for GPS and the three ablation baselines (see Figure 4 in Section 4.3). We also see that the variant mentioned in Section 3.2 (denoted by N-PEPS in the table) performs the best. Note that even though, these results are on test data, we had chosen the best variant based on the results on the validation data.

Table 4: Variation in success ratio for runs across the same machine (run1, run2, run3), different machines (M1, M2, M3) and different test splits (split-1, split-2) for E1

| | split-1 | | | | | | | | | split-2 | | | | | | | | |
| | M1 | | | M2 | | | M3 | | | M1 | | | M2 | | | M3 | | |
| | run-1 | run-2 | run-3 | run-1 | run-2 | run-3 | run-1 | run-2 | run-3 | run-1 | run-2 | run-3 | run-1 | run-2 | run-3 | run-1 | run-2 | run-3 |
|---|---|---|---|---|---|---|---|---|---|---|---|---|---|---|---|---|---|---|
| GPS | 77.4 | 77.4 | 78 | 77 | 77.8 | 77.2 | 77.2 | 77.2 | 77.6 | 78.4 | 78.2 | 78.6 | 78 | 78 | 78 | 78.2 | 78 | 78 |
| Sum | 82.8 | 83.2 | 82.8 | 82.6 | 82.8 | 82.8 | 83.2 | 82.8 | 82.8 | 84.6 | 84.6 | 84.8 | 84.4 | 84.4 | 84.4 | 84.6 | 84.6 | 84.6 |
| Mean | 82.8 | 82.6 | 82.6 | 82.4 | 82.6 | 82.4 | 82.6 | 82.6 | 82.6 | 85 | 85 | 85 | 84.8 | 84.8 | 85 | 85 | 85 | 84.8 |
| Mean+$\mathcal{U}$ | 82.8 | 82.8 | 82.8 | 82.8 | 82.6 | 82.4 | 82.6 | 82.8 | 82.8 | 85 | 85 | 85 | 84.8 | 85 | 84.8 | 85 | 85 | 85 |
| N-PEPS-PP | 86.8 | 86.8 | 86.6 | 86.6 | 86.6 | 86.6 | 86.6 | 86.6 | 86.8 | 89.4 | 89.4 | 89.4 | 89.2 | 89.2 | 89.2 | 89.4 | 89.2 | 89.4 |
| N-PEPS-PP+$\mathcal{U}$ | 86.4 | 86.6 | 86.6 | 86.4 | 86.4 | 86.4 | 86.6 | 86.6 | 86.6 | 88.6 | 88.6 | 88.6 | 88.4 | 88.4 | 88.4 | 88.6 | 88.6 | 88.6 |
| N-PEPS-PG | 86.4 | 86.4 | 86.4 | 86.4 | 86.4 | 86.4 | 86.4 | 86.4 | 86.4 | 87.6 | 87.6 | 87.6 | 87.4 | 87.4 | 87.4 | 87.6 | 87.6 | 87.6 |
| N-PEPS-PG+$\mathcal{U}$ | 87.8 | 88 | 88 | 87.8 | 87.8 | 87.8 | 88 | 88 | 87.8 | 89 | 89 | 89 | 88.8 | 89 | 88.8 | 89 | 89 | 88.8 |

Table 5: Success Ratio with standard error for key variants for E2

| Variant | Length = 5 | Length = 8 | Length = 10 | Length = 12 | Length=14 |
|---|---|---|---|---|---|
| N-PEPS-PG | $78.49 \pm 0.35$ | $45.92 \pm 0.53$ | $31.36 \pm 0.33$ | $22.83 \pm 0.33$ | $17.15 \pm 0.31$ |
| N-PEPS-PG+$\mathcal{U}$ | $78.16 \pm 0.30$ | $46.37 \pm 0.57$ | $31.88 \pm 0.35$ | $23.17 \pm 0.33$ | $17.62 \pm 0.30$ |
| N-PEPS-PP | $78.74 \pm 0.32$ | $45.9 \pm 0.57$ | $31.16 \pm 0.33$ | $22.67 \pm 0.32$ | $16.91 \pm 0.28$ |
| N-PEPS-PP+$\mathcal{U}$ | $78.87 \pm 0.35$ | $44.87 \pm 0.50$ | $30.69 \pm 0.41$ | $22.43 \pm 0.36$ | $16.59 \pm 0.32$ |
| N-PEPS | $79.18 \pm 0.31$ | $\mathbf{47.23 \pm 0.49}$ | $\mathbf{32.3 \pm 0.34}$ | $\mathbf{23.34 \pm 0.28}$ | $\mathbf{17.35 \pm 0.31}$ |
| N-PEPS+$\mathcal{U}$ | $\mathbf{79.19 \pm 0.30}$ | $46.31 \pm 0.61$ | $31.84 \pm 0.36$ | $22.71 \pm 0.28$ | $16.68 \pm 0.21$ |

# F  Additional Results

## F.1  Longer Timeout Results

We wanted to know whether the performance gains of N-PEPS gets translated to scenarios with a higher computational budget (as opposed to a lower budget of 5s in our setting). We performed inference with a timeout of 1000s using our previously trained models for GPS and N-PEPS in the E2 setting. For one test split consisting of 500 examples of length=12, the success ratios for GPS and N-PEPS were 54.38% and 57.14%, respectively. As expected, when given a higher budget, the numbers for both methods increase. However, N-PEPS still outperforms GPS. Note that here we chose the inference hyperparameters based on an educated guess, i.e., $\alpha = 0.8$, PEPS timeout = 160s and the time given to the CA module = 200s. The test performance of N-PEPS is likely to increase further if the values of these hyperparameters are selected from the validation set. This result provides promising evidence towards the wide applicability of our framework for longer timeout settings.

## F.2  Intent Generalization Results

There is an interesting scenario of *intent generalization* where generalization to examples outside of those given as specification is required, in assumption that the additional examples sufficiently define the intent. To see how N-PEPS fares in this setting, we performed experiments where we generated 5 additional IO examples apart from the 5 already present as part of our test data and then evaluated whether the discovered global solutions satisfy the newly generated examples. In Table 6 we provide the success ratio with standard error for GPS and N-PEPS across 30 test splits. As can be seen from the results that even though the numbers have reduced from those provided in the tables provided in Figures 3 and 4 of our paper (as expected because the examples are outside of the specification), N-PEPS still outperforms GPS in both E1 and E2 across all lengths.

Table 6: Success Ratio with standard error for intent generalization experiments

| Method | Length = 4 (E1) | Length = 5 (E2) | Length = 8 (E2) | Length = 10 (E2) | Length = 12 (E2) | Length = 14 (E2) |
|---|---|---|---|---|---|---|
| GPS | $75.80 \pm 0.38$ | $68.31 \pm 0.38$ | $33.87 \pm 0.35$ | $18.19 \pm 0.30$ | $10.99 \pm 0.26$ | $7.48 \pm 0.17$ |
| N-PEPS | $84.09 \pm 0.27$ | $76.16 \pm 0.32$ | $36.33 \pm 0.43$ | $21.02 \pm 0.29$ | $13.17 \pm 0.25$ | $9.17 \pm 0.23$ |

## F.3  Function wise performance

We wanted to see which instructions in the DSL are "difficult" and compare the difficulty across GPS and N-PEPS. To do this, we record the count of instructions in the cases where the model was not

able to find any solution divided by the total count of the instructions. Note that we look only at the operator and not the full statement, i.e., we ignore the arguments. Figure 2 shows this plot for GPS and N-PEPS+$\mathcal{U}$ for E1 with numbers across all 30 test splits. We see that usually higher-order functions like `COUNT, ZIPWITH` are "difficult" and functions like `MAXIMUM, MINIMUM` are comparatively "easy". Also, when compared with GPS, PEPS improves the failure rate across all instructions with improvements ranging from 32.67% for `SUM` to 52.72% for `FILTER`. Other notable improvements being 49.10% for `MAXIMUM`, 44.69% for `MAP`, 45.67% for `SCAN1L` and 47.14% for `TAIL`.

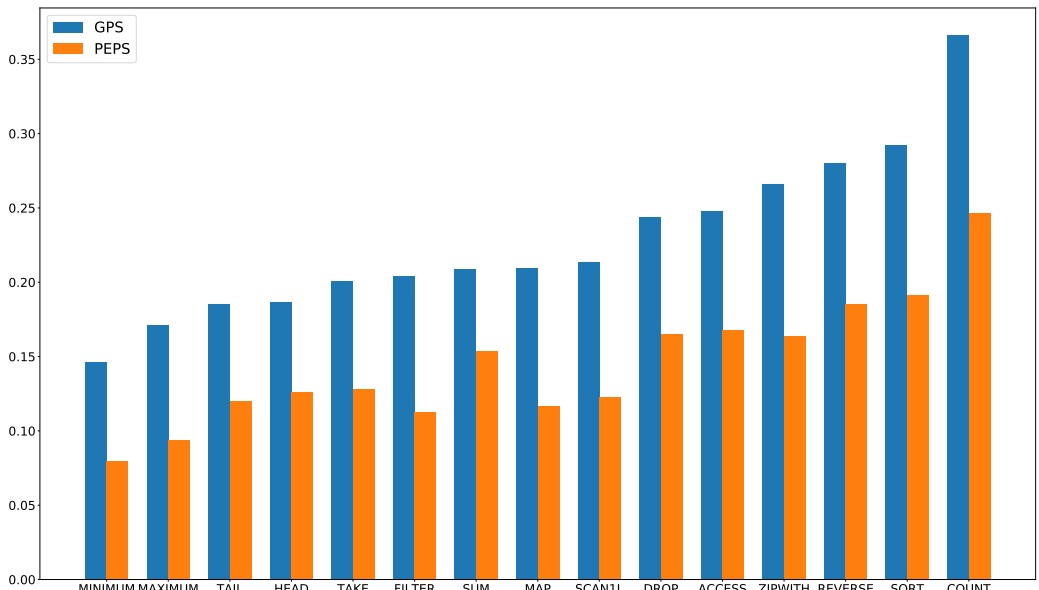

Figure 2: Function-wise breakdown of failing cases for GPS and our N-PEPS model on E1

## F.4 Perfect PE solutions

One of the advantages of PEPS is that we may get a single PE solution which satisfies all IO examples (we call this perfect PE solution). In these cases, we do not even need to go to the CA and depending on when this perfect PE solution is discovered, it can lead to significant time savings (e.g., if the first PE solution discovered turns out to be a perfect solution, then the time taken to find the solution is equal to just the PEPS timeout which is upper bounded by 1/5th of the total timeout). Figure 3 shows the fraction of perfect PE solutions with the length of test programs for N-PEPS for E2. We see that as the program length increases, we have less chances to find a perfect PE solution. This is expected because it will be difficult for a single PE solution to satisfy all IO examples as the programs become lengthy (and hence complex). Note that even though we increase the depth of beam search based on the length of the test program, the overall budget (=5s) and the PEPS timeout (=0.8s in this case) remains the same across different lengths. This also means that for higher lengths, N-PEPS needs to rely more on CA to find a global solution.

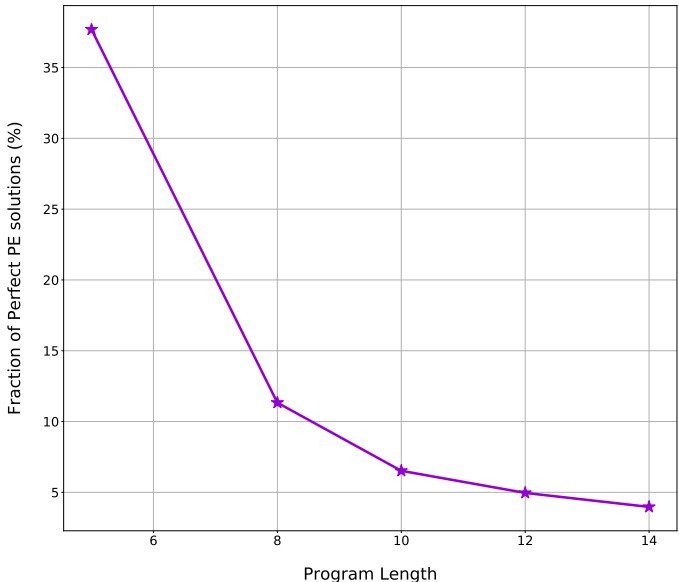

Figure 3: Fraction of perfect PE solutions with length of test programs for our best model for E2

## G More insights into the workings of CA

We tried to gain more insights into how our Cross Aggregator mechanism works. First, we looked into some general patterns learnt by the CA module. Second, we were interested in finding out how often does CA rely on copying from the PE solutions, how often does it generate new operators (in isolation with the GPS module) and how does it generate new operators. The last question is important as it helps us understand the generalization capabilities of CA outside the statements in the PE solutions. Even though we found it difficult to figure out a fixed scheme that worked across all the settings and different examples, by doing nearest-neighbour analysis, we were able to find some useful patterns that might shed some light towards answering this question.

### G.1 General Patterns Learnt by CA

To look for general patterns, we inspected the representations of the final linear layer of our trained CA model (that is responsible for providing the logits used in the global statement prediction). The size of this weight matrix is $n_s \times Z$, where the $i$-th row can be interpreted as a learned embedding corresponding to the statement index $i$. We ran t-SNE on these embeddings and looked for interesting clusters. We found many cases where functionally similar statements or statements with similar signatures were clustered together. We give few examples of these patterns below:

1. `REVERSE b` almost overlaps with `SORT b`. This is interesting because both take in the list `b` and return another list without performing transformations on the elements in `b`.
2. `MINIMUM b`, `MAXIMUM b`, `HEAD b`, and `TAIL b` are clustered together. This is interesting because all these operators select a single element from `b`.
3. `FILTER (ODD) a` is close to `FILTER (ODD) b`. In this case, there is a difference of only the argument. For cases, where the prior statements in the program lead to transformations such that the contents of lists a and b are the same, like `b = SORT a` or `b = REVERSE a`, swapping `FILTER (ODD) a` with `FILTER(ODD) b` and vice-versa will give the same result.

### G.2 Overlap of PE Solutions with Global Solution

We wanted to see in how many cases do the operators present in the global solution also occur in one of the discovered PE solutions. This number gives us a rough estimate of how much can our

attention mechanism do with just trying to copy these operators from one of the PE solutions when synthesizing the global solution. This is a rough estimate because we measure only the overlap of the operators and not statements, i.e., the arguments to the operators in the PE solutions and the global solution can be completely different. Specifically, we record the number of operators that overlap between the global solution (taken as the ground-truth program) and one of the PE solutions, divided by the total number of statements in the ground truth programs across all cases.

The left part of Figure 4 shows the variation of this number with different lengths of test programs for N-PEPS for E2 when $\alpha = 0.8$ (which is the best-chosen hyperparameter value for this setting). We see that there is significant overlap between the operators indicating the quality of our PE solutions. However, the overlap decreases with length, which is also indicated by a decrease in overall success ratio with length (see left part of Figure 4). This is expected because we keep the same budget (PEPS timeout = 0.8s in this case) to discover PE solutions across all lengths. Improvements in the CA architecture focused to improve performance across longer length of programs in limited time budget, can be one of the potential directions to address this.

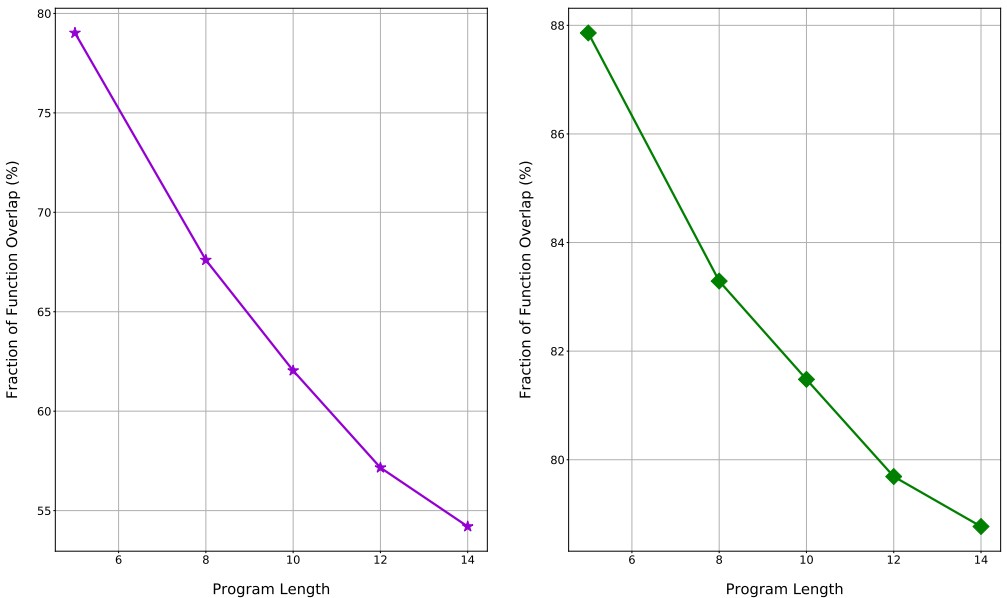

Figure 4: **Variation of fraction of operator overlap with length of test programs for our best model for E2**: *(Left)* $\alpha < 1.0$ (CA + GPS); *(Right)* $\alpha = 1.0$ (CA alone)

There is significant overlap (about 79% for length 5), but not 100% between the operators, highlighting that in many cases (21% for length 5), N-PEPS performs discovery of new operators that are not present in the global solution. To further segregate the role of CA alone in the discovery of new operators as opposed to CA + GPS, we set $\alpha = 1.0$ and analyzed the operator overlap. The right part of Figure 4 shows this variation. As expected, the overlap percentages increase as compared to the case when $\alpha < 1.0$. However, we can see that even when all the contribution to the global solution comes from the CA module alone, there is not a 100% overlap between the operators and therefore, there are non-zero chances of discovery of new operators. From these plots, we can conclude that the CA is not merely a copy mechanism and is useful in scenarios where the discovered PE solutions are not significantly overlapping with the global solution.

### G.3  Sample cases where new operators are discovered

Apart from the analysis done above, we wanted to gain further intuition of how the new operators are being discovered by the CA module. To this effect, we looked at sample cases of the generation of new operators by just CA (i.e., $\alpha = 1.0$). In each box below (Figures 5-9), for test programs of length = 5, we report a case from the cases when the number of new operators discovered is 1, 2, 3, 4 and 5, respectively. The reported example shows the global solution discovered along with the

corresponding PE solutions and is randomly chosen out of the total cases that fall within that category (i.e., not cherry-picked). For clarity, we bold the new operator in the global solution.

**PE Solutions:**

$p_1$ :
a ← LIST
b ← SCAN1L (+) a
c ← MAP (+1) b
d ← SORT c
e ← SCAN1L (-) d

**Global Solution:**

a ← LIST
b ← SORT a
c ← SCAN1L (+) b
d ← MAP (+1) c
e ← **REVERSE** d
f ← SCAN1L (-) e

$p_2$ :
a ← LIST
b ← SCAN1L (+) a
c ← MAP (+1) b
d ← SCAN1L (-) c

$p_4$ :
a ← LIST
b ← MAP (+1) a
c ← SCAN1L (+) b
d ← SORT c
e ← MAP (-1) d

$p_3 = p_5$ :
FAILED

Figure 5: New operators discovered = 1, Total cases = 2169

**PE Solutions:**

$p_1$ :
a ← LIST
b ← FILTER (EVEN) a
c ← MAXIMUM b

$p_2$ :
a ← LIST
b ← MINIMUM a
c ← DROP b a
d ← COUNT (>0) c
e ← ACCESS d a

**Global Solution:**

a ← LIST
b ← MINIMUM a
c ← DROP b a
d ← **SORT** c
e ← **TAKE** b d
f ← MAXIMUM e

$p_3$ :
a ← LIST
b ← MINIMUM a
c ← DROP b a
d ← FILTER (ODD) c
e ← MAXIMUM d

$p_4$ :
a ← LIST
b ← MINIMUM a
c ← DROP b a
d ← FILTER (EVEN) c
e ← MAXIMUM d

Figure 6: New operators discovered = 2, Total cases = 1155

```
                                      PE Solutions:

                                      p₁ :
                                      a ← LIST
                                      b ← MINIMUM a
Global Solution:                      c ← REVERSE a
                                      d ← FILTER (ODD) c
a ← LIST
b ← REVERSE a                         p₂ = p₃ = p₄ :
c ← COUNT (>0) b                      a ← LIST
d ← ACCESS c b                        b ← FILTER (ODD) a
e ← TAKE d b                          c ← REVERSE b
f ← FILTER (ODD) e
                                      p₅ :
                                      a ← LIST
                                      b ← FILTER (<0) a
```

Figure 7: New operators discovered = 3, Total cases = 395

```
                                      PE Solutions:

Global Solution:                      p₁ :
                                      a ← LIST
a ← LIST                              b ← LIST
b ← LIST                              c ← MAP (+1) b
c ← TAIL b                            d ← TAIL c
d ← COUNT (>0) a
e ← DROP d b                          p₂ :
f ← SORT e                            a ← LIST
g ← ACCESS c f                        b ← LIST
                                      c ← TAIL b
```

Figure 8: New operators discovered = 4, Total cases = 119

```
Global Solution:

a ← LIST
b ← LIST                    PE Solutions:
c ← SORT b
d ← MAP (+1) a              p₁ = p₂ = p₃ = p₄ = p₅ :
e ← FILTER (>0) d           FAILED
f ← REVERSE c
g ← ZIPWITH (+) f e
```

Figure 9: New operators discovered = 5, Total cases = 15

Looking at the above samples, there appears to be a trend where discovering fewer and shorter PE solutions leads to more new operators discovered. This may be attributed to the fact that when there is less information in the PE solutions, there is usually more of a need to generate new operators. The example in Figure 9 is an extreme case of this, where no PE solutions were found, so all the operators need to be new.

## G.4   Nearest-neighbour analysis for new operators

To gain intuition about how new operators are being generated by then CA module, we make two assumptions:

- If a statement occurs frequently among the PE solutions, there is a high likelihood that it will also be present in the global solution. We find some evidence of this from our experiments in the paper where we show that the Sum-PEPS baseline performs better than GPS.
- If two statements $s_1$ and $s_2$ are close to each other in the output embedding space (with embeddings $e_1$ and $e_2$), they will also be similar in their corresponding logits. Here, we are assuming that $e_1 \approx e_2 \rightarrow x * e_1 \approx x * e_2$, with $x$ being the input activation.

With the above assumptions, for each of the examples provided in Appendix G.3, we calculated the top-10 nearest neighbours of the PE statements (using the representations obtained in a way described Appendix G.1). After this, we checked if the new operators in the global solution are present as part of the nearest neighbours of the PE statements. The presence of new operators points to a high likelihood of these being ranked higher in the beam search and hence being present in the global solution. In our analysis based on cases provided in Appendix G.3, we did observe this trend. We provide some instances below:

- In Figure 5 above, the statements containing the new operator `REVERSE` occur as the topmost neighbour (based on distance) of `SORT c`, `MAP (+1) b`, as well as among top-3 neighbours of `SCAN1L (+) a`. Note that the variation in certain cases from the general pattern observed before might be attributed to the two assumptions mentioned above not completely holding true in all cases.
  - Top-3 neighbours of `SORT c` (occurs in $p_1, p_4$): [ `REVERSE c, MAP (+1) c, COUNT (>0) c` ]
  - Top-3 neighbours of `MAP (+1) b` (occurs in $p_1, p_2$): [ `REVERSE b, SORT b, COUNT (>0) b` ]
  - Top-3 neighbours of `SCAN1L (+1) a` (occurs in $p_1, p_2$): [ `SCAN1L (-) a, SUM a, REVERSE a` ]
- In Figure 6 above, new operator `SORT` is among the top-2 neighbours of `COUNT (>0) c`. Similarly, the new operator `TAKE` is among the top-2 neighbors of `DROP b a`.
  - Top-3 neighbours of `COUNT (>0) c` (occurs in $p_2$): [ `REVERSE c, SORT c, MAXIMUM c` ]
  - Top-5 neighbours of `DROP b a` (occurs in $p_2, p_3, p_4$): [ `ACCESS b a, DROP b c, DROP b d, DROP c a, TAKE b a` ]