# OpenReview forum: "Learning to Combine Per-Example Solutions for Neural Program Synthesis"
_NeurIPS.cc/2021/Conference — NeurIPS 2021 Poster_

### Official Review · Reviewer_Jkzr · 2021-06-27

**Rating:** 6
**Confidence:** 4

**Summary:**

This paper addresses the problem of learning to synthesize a program that describes a sequence of array operations to realize the transformation between given input/output (I/O) examples. While most existing learning frameworks are designed to find a program that satisfies all I/O examples at once, sometimes it can be inefficient to consider all the examples at every iteration. This paper aims to improve the efficiency of a synthesizing program by aggregating multiple programs that are separately synthesized considering only one example. Specifically, the paper proposes a framework that first (1) finds N per-example programs where each of them satisfies only one of the N examples, and then (2) yields a program that satisfies all examples by combining all per-example programs using a proposed neural network with a multi-head cross-attention mechanism. The experiments show that the proposed framework outperforms a baseline (PCCoder) that considers all examples at once when given a tight time budget. Ablation studies verify the effectiveness of the proposed cross attention scheme. I believe this work studies a promising problem -- building a more efficient neural program synthesis framework and proposes an effective framework to address the problem with sufficient evaluation. Yet, I still have some concerns such as the limited scope and design choices (see below). Currently, I am slightly leaning toward accepting this paper and look forward to the authors' rebuttal.

**Ethical Concerns:**

I currently do not recognize any potential negative impact or ethical concerns for this work.

**Limitations And Societal Impact:**

Described in the main review section.

**Main Review:**

## Paper strengths and contributions

**Motivation and intuition**
The motivation for devising a more efficient neural program synthesis framework is convincing.

**Novelty**
To the best of my knowledge, synthesizing per-example programs first and then learning to aggregate them to form a global solution is novel in the field of neural program synthesis.

**Technical contribution**
- The proposed multi-head cross-attention mechanism for aggregating per-example programs seems effective.
- This work provides insights on neural program synthesis methods working under a limited time budget.

**Parallel execution**
While the proposed framework considers the total time consumed as $\sum_{i}\  the\ time\ of\ finding\ a\ program\ for\ the\ i\text{-}th\ I/O\ pair + the\ time\ of\ aggregating\ them\ programs$, I believe the proposed framework can potentially be extended to even reduce the running time via parallel execution. If I understand correctly, the process of finding per-example programs can run in parallel (while some per-example programs can be applied to more than one I/O pair) so that the execution time could be the summation of $\max_{i}\ the\ time\ of\ finding\ a\ program\ for\ the\ i\text{-}th\ I/O\ pair + the\ time\ of\ aggregating\ all\ programs$.

**Clarity**
The overall writing is clear. The authors utilize figures well to illustrate the ideas.

**Initial experiment**
The goal of the initial experiment is to analyze how per-example programs perform compared to a solution found by a global program synthesis baseline. I like the concept of this experiment, which aims to motivate the proposed framework, while I am not fully convinced by the way how results are interpreted (see below).

**Ablation study**
- Different variations of aggregating per-example programs such as taking summation or mean are evaluated (the table in Figure 3), suggesting the effectiveness of the proposed cross attention mechanism.
- The trade-off analysis on dividing the total timeout into the time given to find PE solutions and the time given to the CA module is informative. This result seems to suggest that the CA module does not need much time. It seems that 5-0.8*5=1 sec for the CA module yields the best performance, which is surprisingly low.
- Different variations of composing the keys by varying the set S in the execution tuple against which per-example programs are executed are evaluated, verifying the effectiveness of the proposed method.

**Experimental results**
- The presentation of the experimental results is clear and easy to follow.
- The experimental results in the tables in Figure 3 and Figure 4demonstrate that the proposed framework outperforms a global program synthesis baseline (PCCoder) in terms of success ratio on test samples with varying lengths when the timeout is set to 5 seconds.
- The success ratio vs. time taken result shown in Figure 3 shows that the proposed framework consistently outperforms PCCoder when taking more than 1 second of computation.
- The attention visualization provides some insights into what the CA module learns.

**Reproducibility**
The code is provided, which helps understand the details of the proposed framework.

## Paper weaknesses & questions

**Limited scope**
While the proposed framework is interesting and inspiring, the evaluations are only limited to array transformation programs. I believe this paper would be much stronger and more convincing if the proposed framework was also evaluated in other popular program synthesis domains such as Karel and string transformation.

**Clarity**
The writing of the method section is not easy to follow. I feel the authors try too hard to be precise by explaining all the details including the dimensions of all the representations and introduce too many notations, which actually stops the flow and makes it hard to read. Revising the section by balancing between giving high-level intuitions and important details would make it easier to follow.

**Pairwise aggregation**
This work proposes to aggregate all the per-example programs at once. I wonder if it is possible to instead perform a sequence of pairwise aggregation to recursively produce the final global solution. It is also possible that the pairwise aggregation can run in parallel to save time. This also aligns well with the high-level idea of this work -- conquering sub-problems first for better efficiency.

**Multiple blocks of CA**
The proposed framework work employs only one block of CA. As models with stacked attention modules have achieved exciting results in a wide range of domains, it would be interesting to see if stacking multiple blocks of CA would yield superior performance even it needs to take some extra time to compute, especially since CA does not need much time it seems (from Figure 4).

**Programs with branching conditions and loops**
It seems to me that both PCCoder and the proposed framework can only synthesize programs consisting of a seq of operations. Yet, programs for many real-world applications often include branching conditions (e.g. if-else) and loops (e.g. while, for). I would like to hear the authors' opinion on how it can be extended to deal with those.

**Combing the predicted statements of GPS and PEPS**
I wonder what the intuition is for combining the predicted statements from both GPS and PEPS. It seems that, from Figure 4 bottom right curve, setting $\alpha=1$ yields the best result, meaning that completely using the predicted statements from PEPS is enough.

**Longer timeout**
While there are definitely some applications that require methods with a tight time budget, there are still other applications that value accuracy more than efficiency. I would like to know how the proposed framework performs with a much longer timeout such as 1000 seconds compared to PCCoder.

**Initial experiment**
While I like the idea of employing this initial experiment to motivate the proposed framework, I am not entirely convinced by the way the result is interpreted. I believe tot(5) and GPS are not directly comparable. Imagine we are solving a hard optimization problem with many constraints. Saying tot(5) (i.e. solve the optimization with some constraints removed) can solve the optimization problem in more cases compared to GPS (i.e. actually solve the problem) does not make much sense because solutions found by tot(5) might not even be meaningful at all. That being said, tot(5) does not tell anything how possible the PE solutions can be combined.

**Supervision of PE**
The paper states "we use $p_g$ as a proxy for ground-truth PE solution".
Isn't this actually "wrong" supervision since $p_g$ might contain some redundant operations that do not need to be applied to some "simpler" I/O pairs?

**Missing related works**
I believe some prior works can be incorporated to make the related work section more comprehensive:
- Synthetic Datasets for Neural Program Synthesis
- Improving Neural Program Synthesis with Inferred Execution Traces
- Neural Program Synthesis from Diverse Demonstration Videos
- Neural Scene De-rendering
- Learning to Describe Scenes with Programs
- Learning to Infer and Execute 3D Shape Programs

**Minor bugs in code**
It is great to include the code so that reviewers can understand the paper better even by just tracing the code. However, it is also encouraged to make sure the code is runnable so that the results can be reproduced, The provided code has several bugs and it took me a long while to make it runnable. These are just some of the minor bugs that I ran into when trying to reproduce the results (I lost track in the middle because there are many):
- missing dependencies: numpy
- Code/scripts/solve_problems.py line 24: a line break is missing
- Code/scripts/solve_problems.py line 269: duplicated `help`
- Code/scripts/gen_train_programs does not run with line 264 `range(min_len, args.max_train_len + 1)` and only works when setting it to start from `min_len+1`. It can't generate program with length equal to zero.
- Code/scripts/gen_train_programs.py line 294: should also `mkdir` when the dir does not exist
- Code/scripts/gen_agg_instances.py: inconsistent use of tabs and spaces in indentation


## Other metrics

### Relevance and significance
- Reasonable contribution to a minor problem

### Novelty
- One idea that surprised me by its originality, solid contributions otherwise

### Technical quality
- Technically adequate for its area, solid results

### Experimental evaluation
- Sufficient evaluation w.r.t. most criteria

### Clarity
- It is mostly clear, but improvements are needed, as recommended in the detailed comments.

**Time Spent Reviewing:**

17

---

> ### Author Response · Authors · 2021-08-10
> **Results on longer timeout, word on parallel execution, applicability to other domains and design choices**
>
> First of all, we are extremely grateful to the reviewer that they invested so much of their valuable time in reading and providing feedback for our paper. Your detailed review with different sections and insightful comments were very useful for us. Thanks a lot for that! We respond to the points raised by you below:
>
> **Parallel execution**: Your understanding is absolutely correct! Indeed we can look for PE solutions in parallel, thereby increasing the speedup in a way that you suggested. However, one could argue that PCCoder can also employ more threads in parallel to speed up their search. Therefore, for a fair comparison with PCCoder, we decided to find PE solutions sequentially where we evaluate both N-PEPS and PCCoder on a single CPU thread (with no parallel computations). Having said that, when being deployed for an application where comparisons with other methods are not required, N-PEPS can significantly boost the speed up by searching for PE solutions in parallel.
>
> **Limited Scope and Applicability to programs with conditionals**: Please see the section on Summary of Reviews and Responses above where we emphasize how the idea of N-PEPS is generally applicable to a wide range of domains as well as provide details of how N-PEPS can be used in conjunction with two different program synthesis frameworks working in the Karel domain (containing loops and conditionals) and the string transformation domain, respectively.
>
> **Multiple blocks of CA**: Indeed as you pointed out, stacking of multiple blocks of CA may lead to improved performance based on the results of transformers in other application areas. As stated in L 190 of the paper, we kept it simple with just one block because we were operating on a low budget (5s). However, when the purpose is not to compare against PCCoder and it is deployed for an application, as you mentioned, we can find PE solutions in parallel and use the saved time for the forward pass over multiple blocks of CA.
>
> **Pairwise aggregation**: This is an interesting suggestion and definitely we will be curious to try this out. Finding PE solutions is just one way of breaking into sub-problems. Since it is the simplest setting, we decided to start with that. Also, we decided to use a cross-aggregator type mechanism that can automatically learn to choose which PE solutions it wants to aggregate. Depending on the budget, as opposed to doing this in one pass though, iteratively performing pairwise or k-way wise aggregation (where k can be learnt) might be useful. Iterative refinement has shown promise in other works of program synthesis like PROBE[4] and we believe it might benefit here as well.
>
> **Clarity on results of Section 4.1**: We agree with all the points you have raised here. In fact, on L 264-265 of the paper, we say that tot(5) is not directly comparable to GPS. We also say in L 270, that the conclusion to draw from this initial experiment is just that there is some potential in thinking of an architecture that can learn to combine the PE solutions. We were very careful not to conclude anything about how they can be combined or even if they are combined they will outperform GPS ( which it did eventually, but not at this stage of the experiments). The main purpose of the initial experiment as stated in L265-272, was to confirm that it is indeed easy to find solutions that satisfy examples partially and make a case for aggregating the PE solutions that can make use of the rich partial clues. We will try to improve the writing to make our points more clear and concise.
>
> **Supervision of PE**: Since we lacked supervision for PE solutions, we decided to use $p_g$ as proxy supervision. Another way of creating this supervision would have been to perform separate PE searches for each example and recording the discovered PE solution as ground-truth. However, this procedure would have required the selection of a specific PE timeout. We didn’t have any good idea of how to select this value as it would have influenced the generated PE solutions, hence the supervision itself. Also, we didn’t know what would have been the best supervision to use for cases where the PE search fails to find a solution. We believe that using $p_g$ as proxy supervision even though not being entirely correct, forces the PE search component to avoid overfitting to a single example and hence is more likely to produce PE solutions that exhibit intent generalization ( generalization to examples outside the ones given as specification).
>
> **Combining GPS and PEPS**: Figure 4 shows the results for the validation data for the E2 setting. Even in that, we have the same validation performance at $\alpha=0.8$ and $\alpha=1.0$. Based on random selection between $\alpha=1.0$ and $\alpha=0.8$, we ended up choosing the latter as the optimal value in this setting. For E1 setting and other N-PEPS variants, we find that a value of $\alpha$ other than 1 is the best. Please see Table 3 in Appendix C.4 for complete details of best hyperparameter values including $\alpha$ for all methods and settings.
>
> **Longer timeout results**: We performed inference with a time budget of 1000s using our previously trained models for GPS and N-PEPS in the E2 setting. For one test split consisting of 500 examples of length=12, the success ratios for GPS and N-PEPS were 54.38% and 57.14%, respectively. As expected, when given a higher budget, the numbers for both methods increase. However, N-PEPS still outperforms PCCoder. Note that here we chose the inference hyperparameters based on an educated guess (due to lack of time), i.e., $\alpha= 0.8$ and PEPS timeout = 160s and time given to the CA module = 200s. The test performance of N-PEPS is likely to increase further if the values of these hyperparameters are selected from the validation set. This result provides promising evidence towards the wide applicability of our framework for longer timeout settings.
>
> **Related Work**: Thanks a lot for the suggestions! We will include these in our camera-ready version upon acceptance.
>
> Hopefully, we have answered all your questions and concerns. If the reviewer agrees that our responses addresses their concerns, we will request them to consider increasing your score. We are happy to answer any further questions during the discussion phase. Thanks again for your detailed review!

---

> > ### Comment · Reviewer_Jkzr · 2021-08-13
> > **Re: Results on longer timeout, word on parallel execution, applicability to other domains and design choices**
> >
> > Thanks for the detailed response.
> >
> > Many of my questions have been addressed. A few points to add:
> >
> > **Limited Scope and Applicability to programs with conditionals**: I still do not think the authors should claim that the proposed method can apply to a wider range of domains than what they have shown in the paper. I would suggest the authors either empirically show it or tone it down.
> >
> > **Supervision of PE**: the response makes sense. I suggest the authors include this discussion in the paper to make it clear.
> >
> > **Longer timeout results**: I am actually surprised that the proposed method (which is more like a "heuristic" - in a good way) still works better than GPS, which is designed to find a program that works for all the examples.
> >
> > For now, I have decided to keep my rating unchanged given that most of my original understanding of the paper is accurate.

---

> > > ### Author Response · Authors · 2021-08-13
> > > **Response to your follow-up**
> > >
> > > Thanks a lot for writing back to us! We are glad we were able to answer your questions. We provide responses to your comments below:
> > >
> > > **Limited Scope and Applicability to programs with conditionals**: We apologize for wording our response to make it sound like a claim we are making. That was not our intention and we will tone it down.
> > >
> > > **Supervision of PE**: That’s a fair point. We will definitely include this discussion in the paper. Thanks for the suggestion!
> > >
> > > **Longer timeout results**: Just to make sure we're on the same page, the cross-aggregator is trained with the same objective as GPS (L 211-215), which is to maximize the likelihood of the global program that satisfies all of the examples. So we view both methods as being designed to find a program that works for all the examples.

---

> > > > ### Comment · Reviewer_Jkzr · 2021-08-26
> > > > **Re: Response to your follow-up**
> > > >
> > > > I see. Thanks for the explanation (and sorry for the delayed response. I actually thought I did reply earlier).

---

### Official Review · Reviewer_syaz · 2021-07-14

**Rating:** 7
**Confidence:** 4

**Summary:**

This paper proposes a new approach for neural program synthesis by first searching for per-example solutions and then learn to combine them into a global solution. The paper uses PCCoder for synthesizing the per-example solutions, and uses an aggregation model based on cross-attention mechanisms. The aggregation model attends to the partially executed program states by the per-example programs in order to choose the next program token. The proposed technique is evaluated on a DSL involving computation on integers and list of integers, and is shown to achieve better performance than PCCoder.

**Limitations And Societal Impact:**

The limitations and potential societal impacts are addressed in the conclusions.


**Main Review:**

Overall, the idea of first synthesizing per-example programs and then combining them in neural program synthesis is interesting and worth exploring. The paper considers and evaluates many different design choices in their approach, which is a plus. The writing of the paper is clear and easy to follow (except too many abbreviations). However, I have some concerns on how generally applicable the proposed approach is, therefore its significance. Please see the following questions for more details.

Questions for authors:
- It seems that the way the proposed aggregation model operates is by choosing one of the next operators from the per-example programs as the next operator in the global program. This can only generate global programs that are stitches of the per-example programs. For a lot of domains this is not likely to work, and more complex ways to join the per-example solutions might be necessary (e.g. adding conditionals). How often does the proposed method generate mere stitches of the PE programs, and how often does it generate programs that contains new operators?

- Is there any examples of non-trivial combination of PE solutions performed by the proposed approach? It's not clear whether the current approach is going to be helpful for other domains where non-trivial combination is required. So experiment on another domain would also be helpful.

- With alpha being a hyperparameter, the proposed approach is using global program search (GPS) as part of itself. If we fix alpha=1, what are the results over all evaluation settings? This way, the comparison will be cleaner.

- What is the takeaway from the N-PEPS variants? They all seems to have a similar performance. Which option does the author recommend?

- The authors might want to include a discussion on the literature on version space algebra used in program synthesis in the related work, since that approach is on combining per-example solutions to a global solution.



**Time Spent Reviewing:**

4.5

---

> ### Author Response · Authors · 2021-08-10
> **Frequency of generation of program stitches, applicability to other domains, performance at $\alpha=1$, best N-PEPS variant and others**
>
> Thanks a lot for your insightful comments! We are glad you found our idea interesting as well as found our evaluations detailed and writing easy to follow. Below we address your questions:
>
> **Frequency of generation of program stitches**: This is an interesting question! We performed some analysis experiments in addition to what we already had in the Appendix to answer this question in detail.
>
>   - *For $\alpha < 1.0$:* In Appendix E.3, for N-PEPS in E2 across all test lengths, we measure the fraction of times there is an overlap of operators found in the PE solutions with the unique operators in the global solution (we take the ground truth global solution for cases where N-PEPS fails to discover any global solution) and divide it by the total number of unique statements in the global solution. We plot this number for different lengths of the test data (see right part of Figure 3). There is significant overlap (about 79% for length 5), but not 100% between the operators, highlighting that in many cases (21% for length 5), N-PEPS performs discovery of new operators that are not present in the global solution. The overlap decreases as the length increases. These results are at alpha=0.8 which is the best-chosen hyperparameter value for this setting ( see Table 3 in Appendix C.4 for best inference hyperparameters for all methods). As a specific example of new operator discovery, please see the second sample case illustrated in Appendix F, where MAXIMUM doesn’t occur in any PE solutions and hence is a new operator discovered.
>
> - *For $\alpha=1$:* As asked by you, for segregating the role of CA alone in the discovery of new operators as opposed to CA + GPS, we present the overlap results of N-PEPS-PG+U with $\alpha=1$ across the 30 test splits for E2. For these results, we look at all discovered global solutions (ignoring failed cases) and see how many unique statements in the global solution overlap with any of the PE solutions and divide it by the number of unique statements in the discovered global solution, across all test splits. The results are given below:
>
>     >| Length 	| Overlap (%) 	|
> |:------:	|:-----------:	|
> |    5   	|    87.86    	|
> |    8   	|    83.29    	|
> |   10   	|    81.48    	|
> |   12   	|    79.69    	|
> |   14   	|    78.77    	|
> |        	|             	|
>
>
> As can be seen from the numbers, even when all the contribution to the global solution comes from the CA module alone, there is not a 100% overlap between the operators and therefore, there are non-zero chances of discovery of new operators. Hence, the CA is not merely a copy mechanism, providing usefulness for cases where the discovered PE solutions are not significantly overlapping with the global solution.
>
> **Applicability to other domains:** Please see the section on Summary of Reviews and Responses above where we emphasize how the idea of N-PEPS is generally applicable to a wide range of domains as well as provide details of how N-PEPS can be used in conjunction with two different program synthesis frameworks working in the Karel domain (containing loops and conditionals) and the string transformation domain, respectively.
>
> **Success Ratio at $\alpha=1$:** Since we were not sure which evaluation setting did you mean here specifically, we report performances for two settings:
>
>  - *Performance of N-PEPS at $\alpha=1$ across E1 and E2 for different lengths:* Below, we report the success ratio with standard error at $\alpha=1$ for N-PEPS+U across 30 test splits for E1 and N-PEPS for E2:
>
>     >|                         Test Length 	| Success Ratio (Std Error) 	|
> |-------------------------------------	|:-------------------------:	|
> |                4 (E1)               	|        86.33 (0.26)       	|
> |                5 (E2)               	|        78.77 (0.33)       	|
> |               8 (E2)                	|        47.23(0.51)        	|
> |               10 (E2)               	|        31.61 (0.36)       	|
> |               12 (E2)               	|        22.73 (0.27)       	|
> |               14 (E2)               	|        17.27 (0.24)       	|
>
>
>    We can see that the results at $\alpha=1$ are almost the same as those reported in Figures 3 and 4 of our paper, though there is a slight decrease from the best value in all cases. This is because the optimal value of $\alpha$ selected for both E1 and E2 was 0.8.
>
>  - *Performance of different methods (including the key variants) at $\alpha=1$ for E2:* We take the validation set and for different values of success ratio corresponding to different PEPS timeouts at $\alpha=1$, we report the maximum value. For reference, we also specify the best value of $\alpha$ chosen for that method along with the success ratio at the chosen value.  The best value of alpha chosen for each of the methods under E1 and E2 is given in Table 3 in Appendix C.4.
>
>     >|    Method   	| Success Ratio at $\alpha=1$  	| Chosen value of $\alpha$ 	| Success ratio at the chosen value 	|
> |:-----------:	|:----------------------------:	|:------------------------:	|:---------------------------------:	|
> |   Sum-PEPS  	|              50              	|            0.2           	|                73.89              	|
> |  Mean-PEPS  	|              50              	|            0.2           	|               73.89               	|
> | Mean-PEPS+U 	|              50              	|            0.4           	|               74.11               	|
> |    N-PEPS   	|             78.01            	|            0.8           	|               78.01               	|
> |   N-PEPS+U  	|             76.32            	|            0.9           	|               76.95               	|
> |  N-PEPS-PG  	|             76.43            	|            0.9           	|                76.64              	|
> | N-PEPS-PG+U 	|             76.74            	|            1.0           	|               76.74               	|
> |  N-PEPS-PP  	|             76.11            	|            0.8           	|               76.11               	|
> | N-PEPS-PP+U 	|             76.74            	|            1.0           	|               76.74               	|
>
> **Best N-PEPS variant:** Based on experiments on both E1 and E2 (Variants of K in Section 4.3 (Table 2 in main paper), the key variant based on the execution tuple obtained by executing each PE program against all IO examples individually (described in Section 4.3) is the one that performs the best and we recommend that. We refer to this variant as N-PEPS everywhere in the paper. The reason this variant performs the best is that it has the largest set of keys ( as compared to N-PEPS-PP and N-PEPS-PG described in Section 4.3), thereby contributing to comparatively richer state features that help the CA module.
>
> **Works on version space algebra:** We will include a discussion on relevant works in this space in our related works section.
>
> Hopefully, we have clarified all the questions raised by the reviewer. If the reviewer agrees that our responses addresses their concerns, we will request them to consider increasing your score. We will be happy to answer any additional questions that the reviewer has. Thanks again for your time!

---

> > ### Comment · Reviewer_syaz · 2021-08-13
> > **Thanks for the response, would still like to see some examples**
> >
> > Thank you for the detailed responses and additional experiments performed. I think the additional results addresses my main concerns about the proposed method can only produce simple stitches of the PE solutions. Currently the authors only show the statistics, I would still like to see some examples where CA generates new operators instead of mere copying. I think the examples can give some insights into how the is generating new operators.

---

> > > ### Author Response · Authors · 2021-08-13
> > > **More examples where CA generates new operators**
> > >
> > > Thanks a lot for writing back to us! Apart from the example specified earlier ( second sample case in Appendix F), we provide additional examples of the generation of new operators for N-PEPS with $\alpha=1$ (only CA contributes to new operators) below. In each heading, we report an example showing the global solution discovered along with the corresponding PE solutions, from the cases when the number of new operators discovered is 1, 2, 3, 4 and 5, respectively for programs of length=5. The reported example is randomly chosen out of the total cases that fall within that category (i.e., not cherry-picked). For clarity, we bold the new operator in the global solution ( for some reason COUNT in Examples 3 and 4 and FILTER in Example 5 was not bolded in rendering).
> > >
> > > **1. Length = 5, New operators discovered = 1, Total cases = 2169**
> > >
> > >  - *Global Solution*:
> > >
> > >     a <- LIST
> > >
> > >     b <- SORT a
> > >
> > >     c <- SCAN1L (+) b
> > >
> > >     d <- MAP (+1) c
> > >
> > >     e <- **REVERSE** d
> > >
> > >      f  <- SCAN1L (-) e
> > >
> > >
> > > - *PE Solutions:*
> > >
> > >    - $p_1$:
> > >
> > >         a <- LIST
> > >
> > >         b <- SCAN1L (+) a
> > >
> > >         c <- MAP (+1) b
> > >
> > >         d <- SORT c
> > >
> > >         e <- SCAN1L (-) d
> > >
> > >    - $p_2$:
> > >
> > >         a <- LIST
> > >
> > >         b <- SCAN1L (+) a
> > >
> > >         c <- MAP (+1) b
> > >
> > >         d <- SCAN1L (-) c
> > >
> > >   - $p_3 = p_5$
> > >
> > >        Failed
> > >
> > >    - $p_4$
> > >
> > >        a <- LIST
> > >
> > >        b <- MAP (+1) a
> > >
> > >       c <- SCAN1L (+) b
> > >
> > >       d <- SORT c
> > >
> > >       e <- MAP (-1) d
> > >
> > > **2. Length = 5, New operators discovered = 2, Total cases = 1155**
> > >
> > >  - *Global Solution*:
> > >
> > >     a <- LIST
> > >
> > >     b <- MINIMUM a
> > >
> > >     c <- DROP b a
> > >
> > >     d <- **SORT** c
> > >
> > >     e <- **TAKE** b d
> > >
> > >      f  <- MAXIMUM e
> > >
> > >
> > > - *PE Solutions:*
> > >
> > >    - $p_1$:
> > >
> > >         a <- LIST
> > >
> > >         b <- FILTER (EVEN) a
> > >
> > >         c <- MAXIMUM b
> > >
> > >    - $p_2$:
> > >
> > >         a <- LIST
> > >
> > >         b <- MINIMUM a
> > >
> > >         c <- DROP b a
> > >
> > >         d <- COUNT (>0) c
> > >
> > >         e <- ACCESS d a
> > >
> > >   - $p_3$
> > >
> > >        a <- LIST
> > >
> > >        b <- MINIMUM a
> > >
> > >        c <- DROP b a
> > >
> > >        d <- FILTER (ODD) c
> > >
> > >        e <- MAXIMUM d
> > >
> > >
> > >    - $p_4$
> > >
> > >        a <- LIST
> > >
> > >        b <- MINIMUM a
> > >
> > >        c <- DROP b a
> > >
> > >       d <- FILTER (EVEN) c
> > >
> > >       e <- MAXIMUM d
> > >
> > > **3. Length = 5, New operators discovered = 3, Total cases = 395**
> > >
> > >  - *Global Solution*:
> > >
> > >     a <- LIST
> > >
> > >     b <- REVERSE a
> > >
> > >     c <- **COUNT** (>0) b
> > >
> > >     d <- **ACCESS** c b
> > >
> > >     e <- **TAKE** d b
> > >
> > >      f <- FILTER (ODD) e
> > >
> > >
> > > - *PE Solutions:*
> > >
> > >    - $p_1$:
> > >
> > >         a <- LIST
> > >
> > >         b <- MINIMUM a
> > >
> > >         c <- REVERSE a
> > >
> > >         d <- FILTER (ODD) c
> > >
> > >    - $p_2 = p_3 = p_4$:
> > >
> > >         a <- LIST
> > >
> > >         b <- FILTER (ODD) a
> > >
> > >        c <- REVERSE b
> > >
> > >   - $p_5$
> > >
> > >        a <- LIST
> > >
> > >        b <- FILTER (<0) a
> > >
> > > **4. Length = 5, New operators discovered = 4, Total cases = 119**
> > >
> > >  - *Global Solution*:
> > >
> > >     a <- LIST
> > >
> > >     b <- LIST
> > >
> > >     c <- TAIL b
> > >
> > >     d <- **COUNT** (>0) a
> > >
> > >     e <- **DROP** d b
> > >
> > >     f <- **SORT** e
> > >
> > >     g <- **ACCESS** c f
> > >
> > >
> > > - *PE Solutions:*
> > >
> > >    - $p_1$:
> > >
> > >         a <- LIST
> > >
> > >         b <- LIST
> > >
> > >         c <- MAP (+1) b
> > >
> > >         d <- TAIL c
> > >
> > >    - $p_2$:
> > >
> > >         a <- LIST
> > >
> > >         b <- LIST
> > >
> > >         c <- TAIL b
> > >
> > > **5. Length = 5, New operators discovered = 5, Total cases = 15**
> > >
> > >  - *Global Solution*:
> > >
> > >     a <- LIST
> > >
> > >     b <- LIST
> > >
> > >     c <- **SORT** b
> > >
> > >     d <- **MAP** (+1) a
> > >
> > >     e <- **FILTER** (>0) d
> > >
> > >     f <- **REVERSE** c
> > >
> > >     g <- **ZIPWITH** (+) f e
> > >
> > >
> > > - *PE Solutions:*
> > >
> > >    - $p_1 = p_2 = p_3 = p_4 = p_5$:
> > >
> > >         Failed
> > >
> > > There appears to be a trend where discovering fewer and shorter  PE solutions leads to  more new operators discovered. This may be attributed to the fact that when there is less information in the PE solutions, there is usually more of a need to generate new operators. The last example is an extreme case of this, where no PE solutions were found, so all the operators need to be new.

---

> > > > ### Comment · Reviewer_syaz · 2021-08-14
> > > > **Re: More examples**
> > > >
> > > > Thanks for showing these examples! I think these are interesting, as well as the trend the authors found. From looking at these examples, I wasn't really able to understand how these generated new operators are able to "join" different PE solutions (i.e. make a more generalized solution that works for every case). Do the authors spot any common schemes of how the new operators are making a generalized solution out of the PE solutions?

---

> > > > > ### Author Response · Authors · 2021-08-20
> > > > > **Observations regarding some common patterns in CA**
> > > > >
> > > > > Sorry for the late response! As per our interpretation of your question, you were interested in knowing about some common patterns that the CA has learnt (particularly what is the mechanism it uses to generate new operators). We looked at several samples, but we found it difficult to figure out a fixed scheme that worked across all the settings and different examples. However, we present below some observations that may shed some light toward answering your question.
> > > > >
> > > > > - To look for general patterns, we inspected the representations of the final linear layer of our trained CA model (that is responsible for providing the logits used in the global statement prediction -- please see L 186-187, L 197-198 of our paper). The size of this weight matrix is $n_s \times Z$, where the $i$-th row can be interpreted as a learned embedding corresponding to the statement index $i$. We ran t-SNE on these embeddings and looked for interesting clusters. We found many cases where functionally similar statements or statements with similar signatures were clustered together. We give few examples of these patterns below ( we will include these plots in the updated version of the paper):
> > > > >      - `REVERSE b` almost overlaps with `SORT b`. This is interesting because both take in the list `b` and return another list without performing transformations on the elements in `b`.
> > > > >
> > > > >     - `MINIMUM b`, `MAXIMUM b`, `HEAD b`, and `TAIL b` are clustered together. This is interesting because all these operators select a single element from `b`.
> > > > >
> > > > >     - `FILTER (ODD) a` is close to `FILTER (ODD) b`. In this case, there is a difference of only the argument. For cases, where the prior statements in the program lead to transformations such that the contents of lists `a` and `b` are the same, like `b = SORT a`  or` b = REVERSE a`, swapping `FILTER (ODD) a` with `FILTER(ODD) b` and vice-versa will give the same result.
> > > > >
> > > > >
> > > > > - To gain intuition about how new operators are being generated, we make two assumptions:
> > > > >
> > > > >     - If a statement occurs frequently among the PE solutions, there is a high likelihood that it will also be present in the global solution. We find some evidence of this from our experiments in the paper where we show that the Sum-PEPS baseline performs better than GPS.
> > > > >
> > > > >     - If two statements $s_1$ and $s_2$ are close to each other in the output embedding space (with embeddings $e_1$ and $e_2$), they will also be similar in their corresponding logits. Here, we are assuming that $e_1 \approx e_2 \Rightarrow x * e_1 \approx x * e_2$, with $x$ being the input activation.
> > > > >
> > > > >      With the above assumptions, for each of the examples provided in our earlier response, we calculated the top-10 nearest neighbours of the PE statements (using the representations obtained in a way described in the earlier point). After this, we checked if the new operators in the global solution are present as part of the nearest neighbours of the PE statements. The presence of new operators points to a high likelihood of these being ranked higher in the beam search and hence being present in the global solution. In our analysis based on the earlier examples, we did observe this trend. We provide some instances below:
> > > > >
> > > > >     - In Example 1 above, the statements containing the new operator `REVERSE` occur as the topmost neighbour (based on distance) of `SORT c , MAP (+1) b `, as well as among top-3 neighbours of `SCAN1L (+) a `. Note that the variation in certain cases from the general pattern observed before might be attributed to the two assumptions mentioned above not completely holding true in all cases.
> > > > >
> > > > >       - *Top-3 neighbours of `SORT c`* (occurs in $p_1, p_4$): [`REVERSE c`, `MAP (+1) c`, `COUNT (>0) c`]
> > > > >
> > > > >       - *Top-3 neighbours of `MAP (+1) b`* (occurs in $p_1, p_2$): [`REVERSE b`, `SORT b`, `COUNT (>0) b`]
> > > > >
> > > > >       - *Top-3 neighbours of `SCAN1L (+1) a`* (occurs in $p_1, p_2$): [`SCAN1L (-) a`, `SUM a`, `REVERSE a`]
> > > > >
> > > > >     - In Example 2 above, new operator `SORT` is among the top-2 neighbours of `COUNT (>0) c`. Similarly, the new operator `TAKE` is among the top-2 neighbors of `DROP b a`.
> > > > >
> > > > >       - *Top-3 neighbours of `COUNT (>0) c`* (occurs in $p_2$): [`REVERSE c`, `SORT c`, `MAXIMUM c`]
> > > > >
> > > > >       - *Top-5 neighbours of `DROP b a`* (occurs in $p_2, p_3, p_4$): [`ACCESS b a`, `DROP b c`, `DROP b d`, `DROP c a`, `TAKE b a`]

---

> > > > > > ### Comment · Reviewer_syaz · 2021-08-20
> > > > > > **Great investigation!**
> > > > > >
> > > > > > Thank you for this great investigation! This helps a lot in understanding how the CA is generating new operators. I am a lot more convinced now that the proposed technique is able to learn some interesting ways to combine per example solutions into global solutions.
> > > > > >
> > > > > > The responses from the authors have cleared all my original concerns. The experimental design and analysis are systematic and thorough, which is a very strong point of this paper. I will increase my score.

---

### Official Review · Reviewer_vnpm · 2021-07-17

**Rating:** 7
**Confidence:** 4

**Summary:**

The paper proposes a method for neural synthesis of straight-line programs from I/O examples that works by learning
candidate solutions for individual examples and then combining their instructions into an overall solution using a
separate model. They build upon prior work on the DeepCoder dataset (PCCoder) that remains SOTA for neural-guided search
on these programs, and extend it with a second search procedure to assemble lines from per-example candidate solutions
into an overall solution. The resulting search significantly outperforms the PCCoder baseline with roughly the same loss
of generalization to longer programs.

**Ethical Concerns:**


No issues or concerns.



**Limitations And Societal Impact:**


A brief note on impact is included in Section 6, likely sufficient for the paper's topic area.
I did not find any explicit discussion of limitations anywhere.


**Main Review:**


Strengths:
- An original idea for straight-line synthesis – use programs satisfying individual examples as a vocabulary of
  possibly-useful instructions, and learn to aggregate them into a global solution.
- Strong empirical results with investigation of attention and limited generalization.
- Well-described and rigorous evaluation setup.
- Decent overview of related work.

Weaknesses:
- Some choices in the architecture could be better motivated or investigated. The authors already compare to several
  ablations, but they come too late in Section 4. The paper lacks a high-level description of the current encoding
  process and its explored alternatives somewhere around page 5. Currently, the brief "Motivation" paragraph sets up
  _why_ PEPS looks at per-example program states and lines as clues, but then stops just short of explaining _what_ PEPS
  could do with those states/lines at an equally high level.
- Generalization experiments are lacking. The only one presented is length generalization from 12 to 14 lines, which is
  not even particularly ambitious (of course, that still drops every model's performance by ~26%). In contrast, prior
  work on by-example synthesis such as PCCoder, Flash*, RobustFill is also concerned with example overfitting in
  addition to length or compositional overfitting. Specifically:
  - The best model of PCCoder is trained to embed 10 examples but only presented with 5 examples at test time. How would
    an equivalent version of PEPS's cross-aggregator model perform? As presented, it should be able to aggregate any
    variable number of programs that was found in the first phase, but it learns its cues based on the set of available
    keys (program states/lines), which at training time is induced by available I/O examples. Would a bigger set of keys
    yield a stronger cross-aggregator (thanks to a richer sample of state features) or a weaker one (thanks to reliance
    on the higher chance of better-matching PE program being there)?
  - Flash*, RobustFill, Karel, and many other program synthesis settings value _intent generalization_: whether the
    found program satisfies not only the given examples but also a hidden one (in assumption that the given 5
    sufficiently define the intent). DeepCoder and PCCoder do not measure it, presumably because their search methods
    are not amenable to that objective (the trained models optimize correctness w.r.t. the given 5 examples). But PEPS
    is much more powerful! In a sense, it already optimizes example generalization – the whole point of
    cross-aggregation is to use programs that satisfy _some_ examples as building blocks or cues to the program that
    satisfies _more_ examples. We might discover that this method learns to synthesize a _more generalizable_ program,
    even for examples not shown to the aggregator model. That would be a very significant finding for the field.


## Opportunities for improvement

Related work might discuss connections to learning of decision trees and older set-cover driven techniques to synthesize
if-else chains in programming by examples. They also take programs that satisfy a subset of examples, and delegate
satisfaction of the remaining ones recursively to the program in the "alternative" branch. In contrast to PEPS, they use
all learned programs as independent branches at test time - not an option for this DSL.
1. Alur, R., Radhakrishna, A., & Udupa, A. (2017, April). Scaling enumerative program synthesis via divide and conquer.
   In International Conference on Tools and Algorithms for the Construction and Analysis of Systems (pp. 319-336).
   Springer, Berlin, Heidelberg.
2. FlashFill and FlashMeta: algorithms for learning If/ElseIf/Else chains.

The current matching process is based on similarity of _the current program state_ with _the state of each candidate
program right before the line of interest_. Counter-intuitively, it does not consider whether relying on that line would
bring the program state "closer" to desired output. In fact, the "final output" part of the state seems underutilized
and might even be superfluous. As such, the aggregator network does not rely on instruction _semantics_, only on its
syntax and its proximity to the current state.
Possible concrete suggestions: split Equations 1-3 into two attention blocks, one against keys={states before each line} and one
against keys={states after each line}. Or concatenate them before embedding the key matrix. Or max-pool from each
(before, after) pair to form the key matrix – there are many architectural options.

How often is the synthesized line has the same operator, or even identical to the lines the aggregator attends to? It
seems to be the case for most of the examples. Would a copy mechanism for $o_t$, or even $s_t$ perform just as well?
Even if it makes occasional mistakes, it might cut down inference time a little, which would free up search budget.

## Clarity

Most parts of the paper are clear, and include much-appreciated diagrams for the search process and model pipeline. Some
enhancements:
- High-level description of cross-aggregation and its ablations (see Weaknesses above).
- Usage of the term "GPS" as a substitute for "baseline PCCoder" is confusing, especially in Section 4 when they are
  used interchangeably and inconsistently. There seems to be no need to introduce a new, more general term, if the paper
  explores exactly one baseline instantiation of it.
- Equation 4 does not type-check (at least in my brain).
  First, the predicted statement $\hat{s}^t$ is multiplied by a
  real scalar. Previously, $s^t$ was defined as a one-hot vector (L97), and $\hat{s}^t$ as its estimate (L214). I
  suppose in practice $\hat{s}^t$ likely means predicted logits and thus Equation 4 is meant to be interpreted
  literally, but that was never properly defined in the paper 🙂
  Second, using real-valued subscripts $\alpha$ and $1-\alpha$ to obscurely denote "from cross-aggregation" and "from
  program search" respectively does not make much notational sense.


**Time Spent Reviewing:**

6

---

> ### Author Response · Authors · 2021-08-10
> **Intent generalization experiments, effect of a larger set of keys, CA acting as copy mechanism, improving writing and others**
>
> Thanks a lot for the insightful comments and suggestions for improvements on our paper! We are glad that you liked our idea and were happy with the experimental setup, results and figures :) We respond to the points that you have raised below:
>
> **High-level description of CA**:  Thanks a lot for suggesting this! We will definitely include a brief description of our motivation for using the cross-attention mechanism as a means for aggregating the PE solutions in the camera-ready version. Similarly, we will also include the discussion about the ablation baselines earlier in the paper.
>
> **Generalization**: In order to answer your question about the performance of N-PEPS  for intent generalization setting, we performed experiments where we generated 5 additional IO examples apart from the 5 already present as part of our test data and then evaluated whether the discovered global solutions satisfy the newly generated examples. Below, we provide the success ratio with standard error for PCCoder and N-PEPS across 30 test splits. As can be seen from the results that even though the numbers have reduced from those provided in the tables provided in Figures 3 and 4 of our paper (as expected because the examples are outside of the specification), N-PEPS still outperforms PCCoder in both E1 and E2 across all lengths.
>
> > |  Length 	|     PCCoder     	|    N-PEPS   	|
> |:-------:	|:-----------:	|:-----------:	|
> |  4 (E1) 	| 75.80(0.38) 	| 84.09(0.27) 	|
> |  5 (E2) 	| 68.31(0.38) 	| 76.16(0.32) 	|
> | 8 (E2)  	| 33.87(0.35) 	| 36.33(0.43) 	|
> | 10 (E2) 	| 18.19(0.30) 	| 21.02(0.29) 	|
> | 12 (E2) 	| 10.99(0.26) 	| 13.17(0.25) 	|
> | 14 (E2) 	| 7.48 (0.17) 	|  9.17(0.23) 	|
>
> As you rightly pointed out, out-of-distribution generalization is an important problem to be addressed. We have left extensive evaluation of N-PEPS on other generalization settings as future work. As mentioned in L 377, in particular, we are interested in compositional generalization, where the statements in the test programs have not been seen exactly during training (say `FILTER (lambda x: x >0)`, but their sub-components have been seen (e.g. `FILTER (lambda x: x < 5)`, `MAP (lambda x: x > 0)`, etc.). You said that “ ...PCCoder, Flash*, RobustFill is also concerned with example overfitting in addition to length or compositional overfitting”. Based on our understanding none of these works evaluate compositional generalization or length generalization (except PCCoder that evaluated from lengths 4 to 5 and lengths 12 to 14--we did the latter as well). Did you mean that these works evaluate example overfitting only?
>
> **Effect of more keys**: In our work, we train both the PCCoder and N-PEPS models with 5 examples each and do inference using 5 examples. We assume you are talking about the PCCoder_Ten5 results presented in Table 5 of the original PCCoder paper. This will indeed be an interesting thing to try for the future. Unfortunately, we were not able to show results of this setting for N-PEPS in time for the rebuttal. However, we still had investigated the effect of the size of the key set on the performance of N-PEPS. We test this effect in two ways:
>
>   (a) *Key Variants*: The three variants of K discussed in Section 4.3 lead to different sizes of the set of keys. The N-PEPS version that we use and that performs the best, has the highest number of keys obtained by executing each PE program against all five IO examples individually ( as opposed to N-PEPS-PG where each PE program is executed against the global set and N-PEPS-PP where each PE program is executed against the set of examples that it satisfies).
>
>   (b) *Varying number of programs while training*: As mentioned in Appendix B.2, we have two settings `tot` and `all`. In the `tot` setting, we take only the PE programs that satisfy all examples in total, whereas, in `all` we take all PE programs. Also, we consider three types of PEPS timeouts that essentially lead to different PE programs for the same global program.
>
> Both these settings are treated as hyperparameters for the training of CA module. Table 2 in Appendix C.3 shows the best values of hyperparameters selected for all key variants. Based on the observations above, the answer to your question is that a *bigger set of keys adds to a richer set of state features* as you pointed out, and hence are beneficial.
>
> **Related Work**: Thanks a lot for pointing the papers! We will definitely look into them and cite the ones not cited already, in the camera-ready version of the paper.
>
> **Looking at the after execution state**: Thanks for the excellent suggestion! We will definitely try this out. It will be interesting to see the increase in computational complexity due to added parameters vs the improvement in performance obtained by incorporating the state after execution of a PE statement within the keys of that particular statement. In the current formulation, after execution program state is incorporated, but as part of the keys for the PE statement following the current one.
>
> **How much does the mechanism rely on copying**: This is an excellent question! Please see the section on Summary of Reviews and Responses above where we show results on the amount of overlap between the operators in the PE solutions and the discovered global solution when taking all the contribution from CA ($\alpha=1$) as well as taking the combined contribution from CA and PCCoder ($\alpha< 1$).
>
> **Equation 4 notation**: In L 197-198 we did mention that $\hat{s}^t$ are the logits from a softmax. The use of $\alpha$ and $1-\alpha$ was to emphasize that there is just one hyperparameter whose value needs to be optimized.
>
> **Limitations discussion**: Due to lack of space, we were not able to put a dedicated section for limitations but as mentioned in the checklist, we include some lines towards the end of the first paragraph of Section 5.

---

> > ### Comment · Reviewer_vnpm · 2021-08-20
> > **Thanks**
> >
> > Thanks for the update! The intent generalization experiments look great. Surprisingly, the _difference_ in improvements in this setting is less than the original setting in the paper. (For instance, in the L=14(E2) setting, the original improvement was $14.71 \to 17.35$ and in its harder intent-generalization version it's $7.48 \to 9.17$ – slightly lower delta, consistently across all settings.) I expected the opposite effect intuitively in my comment, hoping that N-PEPS can handle the generalization setting much better. But empirically, the improvement of N-PEPS over PCCoder here is consistent with its improvement over PCCoder in general, so my intuition was wrong.
> >
> > > Did you mean that these works evaluate example overfitting only?
> >
> > Yes, that's what I meant – awkward phrasing. They focus on example generalization wherein other works in the code generation space focus on length or compositional generalization. Either way, all are important settings to investigate.

---

### Author Response · Authors · 2021-08-10
**Summary of Reviews and our Responses**

First of all, we want to thank all the reviewers for their detailed feedback and comments on our draft! We really appreciate the efforts put forth by them, especially Reviewer Jkzr. We are glad that Reviewer vnpm and Reviewer syaz liked the originality and novelty of our idea and Reviewer syaz found our technique interesting. Reviewer vnpm seemed to like our “..strong empirical results” with “..well described and rigorous evaluation setup”, Reviewer syaz appreciated that our paper  “..considers and evaluates many different design choices”  and Reviewer Jkzr liked our experimental results and rigor in evaluations. In general, all the reviewers also seemed to like our presentation of experiments and figures and writing. We also want to thank Reviewer vnpm and Reviewer Jkzr for interesting design suggestions towards the improvement of our framework. The main points raised by the reviewers were questions related to the applicability of our framework to other settings, understanding if our cross-aggregator is working as a copy mechanism and suggestions for improved writing. We address these below:

***1. Applicability of N-PEPS to other settings: New experiments show good results in two requested settings & we provide insights into how N-PEPS could be used with other requested DSLs in the future.***

- **Intent generalization experiments**: Reviewer vnpm was curious to know the performance of N-PEPS in the setting where generalization to examples outside of those given as specification is required. To answer this question, we perform experiments where the discovered global solution is evaluated on 5 additional test examples (see the response to Reviewer vnpm for details of the experiment and results). We find that similar to the results shown in the paper, N-PEPS significantly outperforms PCCoder in both evaluation settings (E1 = trained on programs up to length 4 and evaluated on programs of length 4 as well as E2 = trained on programs up to length 12 and evaluated on programs of length 5, 8, 10, 12 and 14), highlighting the usefulness of N-PEPS to this setting.


- **Longer timeout experiments**: Reviewer Jkzr wanted to know whether the performance gains of N-PEPS get translated to scenarios with a higher computational budget ( as opposed to a lower budget of 5s in our setting). To answer this question, we performed inference with a time budget of 1000s and show the superior performance of N-PEPS (57.14%) as compared to GPS (54.38%) for test programs of length 12. This provides promising evidence towards the wide applicability of our framework for longer timeout settings (see the response to Reviewer Jkzr for details of the experiment).


- **Applicability to other domains**(*modified on August 19th based on Reviewer Jkzr's comments*): Reviewer syaz and Reviewer Jkzr raised concerns over the applicability of N-PEPS to domains other than straight-line code. Since in this work we have built upon PCCoder, it was more natural to show results in the same DSL as they have considered. However, we believe that the notion of breaking into sub-problems to find programs that satisfy a single example and then combining these to a global solution has the potential to be widely applicable and might be used in conjunction with most of the machine learning based program synthesis systems that are applicable to other DSLs.  Below, we give examples of how N-PEPS might be incorporated into two existing program synthesis frameworks that work on different program domains to make our point more concise. Please note that we can’t say with absolute certainty how N-PEPS will be applied to these domains or whether it will perform well there or not. The below discussion is not a claim that we are making as part of our paper. It is just an attempt to address the reviewer's concerns based on our intuitions and understanding.

    - *Program synthesis framework in Chen et. al [6] in **Karel domain** (containing loops and conditionals)*: Instead of feeding all IO examples to the neural network encoder, we can obtain the PE solutions and the PE states (hidden state representation $h$ from the LSTM) by feeding just one example as input (this is done once at the beginning of Algorithm 1 in [6]). The next line of the global solution can then be generated by following the default synthesis procedure (outlined in Algorithm 1 of [6]), except that now the synthesizer ($\Gamma$ in lines 4 and 7 of Algorithm 1) takes in the PE solutions and corresponding PE states as additional inputs, combining the aggregation due to CA and the default neural decoder.

    - *PROBE [4] used in **string transformation domain***: Instead of selecting promising solutions based on heuristics (SELECT step at Line 8 of Algorithm 2 in [4]), we can select solutions that satisfy individual examples. Then, we can aggregate these PE solutions using a modified version of our CA mechanism (by replacing UPDATE step at Line 10 of Algorithm 2).

  One of the advantages of our framework is that based on the domain in question, the value of $\alpha$ may be chosen appropriately to account for the tradeoff between the contribution due to aggregation by CA vs the default search/synthesis component of the program synthesizer used. As mentioned in L 397 of our paper, we have left extensive evaluation on the Karel domain as future work.


- **Experiments on other domains** (*added on August 11th*): Reviewer Jkzr felt that our paper will be more convincing if we had experiments on other domains. We believe that the paper in its current form is significant enough to be accepted because of the following reasons:

    - The straight-line DSL used in this paper has been used in previous related works, DeepCoder[3] and PCCoder[29] that have shown results *only* on this DSL and gotten accepted at ICLR (a machine learning conference at par with the NeurIPS standards) and NeurIPS, respectively. This DSL has been inspired by tasks appearing on real programming competition websites (see Appendix A of DeepCoder[3] for more details), and even though might not be fairly complicated, but we believe is expressive enough to evaluate and serve as proof-of-concept for a new program synthesis framework like ours.

    - Our basic idea of breaking the standard program synthesis pipeline into two stages: (a) discovering PE solutions, and (b) learning to aggregate the PE solutions such that it leads to a global solution, was found interesting and novel by the reviewers. The reviewers also agreed that we have done rigorous evaluations and demonstrated strong empirical results to support our claims. We have done extensive experiments with ablation baselines, multiple test splits, evaluation settings, multiple lengths, and experiments analyzing the workings of our frameworks like attention visualizations, plots with variations of hyperparameters, and key variants etc. in the main paper and appendix. We have also added intent generalization, longer timeout results and additional overlap analysis during the rebuttal. We believe these make a strong case for the robustness and value of our framework.

    We don't see fundamental obstacles towards applying the idea of finding and aggregating per-example solutions in DSLs with loops and conditionals, as explained in the previous point. So while additional experiments are always nice to have, we feel that the current DSL is expressive enough to support the claims that we have made in the paper.

***2. Copying and Role of $\alpha$: CA module alone can lead to the discovery of new operators***
All the reviewers seemed to be interested in knowing that in how many cases are the lines in the global solution mere stitches of the PE solutions and whether CA is only copying the PE solutions. To address this question, we present results measuring the overlap between the operators present in PE solutions and global solution both when $\alpha=1$, i.e., all the contribution comes from CA, and when $\alpha < 1.0$, i.e., contribution comes both from CA and GPS (see the response to Reviewer syaz for details of experiments and results). In both cases, we highlight that N-PEPS leads to the discovery of new operators, and hence is not merely generating stitches of PE solutions.

***3. Improving writing:*** We will include a high-level description of how CA should use the program states and include the works mentioned by reviewers to make the related works section of our paper more comprehensive.

In addition to this, we have responded in detail to specific comments raised by the reviewers in the reviewer-specific responses. We are very happy to respond to additional questions/comments during the discussion period.

---

> ### Comment · Reviewer_Jkzr · 2021-08-13
> **Karel domain**
>
> I am still not sure how the proposed framework can be trivially applied to the Karel domain. I thought PE solutions would just consist of a sequence of action tokens (e.g. move(), turnLeft(), turnRight()) and the CA module needs to somehow produce control flows such as while loop, if-else statement. I do not know how that is possible given the current design of the CA module.

---

> > ### Author Response · Authors · 2021-08-13
> > **Re: Karel domain**
> >
> > We apologize for wording our response to make it sound like a claim we are making. That was not our intention and we will tone it down. We agree with you that it is not trivial to apply the proposed framework to the Karel domain. We can’t say with absolute certainty how N-PEPS will be applied to the Karel domain or whether it will perform well there or not. The below discussion is an attempt to answer your question with our intuitions and understanding.
> >
> > In our view, the generated PE solutions would contain control flows statements such as if-else and while loop. This is because these control flows are part of the Karel DSL itself and any program synthesizer conditioned on the syntax of the DSL will generate programs that contain these control flows. Most importantly, since we are providing $p_g$ as proxy supervision ( the target global program will definitely contain control flow statements) for training the PE search component, we have additional reasons to believe that the PE solutions will contain these control flow statements (this ties back to our point about using $p_g$ as supervision). However, even if we assume hypothetically that somehow the PE solutions do not contain these control flow tokens, we believe that our CA module would be able to generate these “new” control flow tokens because it is explicitly being trained to maximize the likelihood of the global program (L 211-215). We want to emphasize that our model is not restricted to only copying statements present in the PE solutions (please see response to Reviewer syaz where we show statistics and examples where the CA module generates new operators).

---

> > > ### Comment · Reviewer_Jkzr · 2021-08-14
> > > **Re: Re: Karel domain**
> > >
> > > Thanks for the explanation. It sounds a little more convincing to me now. I have no further questions at this point.

---

> > > > ### Author Response · Authors · 2021-08-19
> > > > **Edited response based on your comments**
> > > >
> > > > Based on your comments, we have edited our response under the bullet point *Applicability to other domains* above. Thanks a lot again for your comments!

---

### Decision · Program_Chairs · 2021-09-27

**Decision:**

Accept (Poster)

**Comment:**

The paper presents a new technique for synthesizing straight-line programs from input-output examples. The core novelty in the technique is that rather than trying to synthesize a program that works correctly for all available examples, the algorithm first finds separate programs that work for each individual example and then uses a cross aggregator to merge the individually working programs into a program that works correctly on all inputs.

As the authors point out in their related work section, there is precedent for this idea in the program synthesis community, but this has never been done before in a neural setting.

The authors addressed many of the questions and concerns raised in the reviews and provided a very thorough rebuttal. I think the main open question is the generality of the technique, since it is only evaluated on an important but limited domain, and it is not clear how to generalize it to more complex settings.

That said, this paper presents an interesting and novel contribution and should be accepted.